# PROF: AN LLM-BASED REWARD CODE PREFERENCE OPTIMIZATION FRAMEWORK FOR OFFLINE IMITATION LEARNING

## ABSTRACT

Offline imitation learning (offline IL) enables training effective policies without requiring explicit reward annotations. Recent approaches attempt to estimate rewards for unlabeled datasets using a small set of expert demonstrations. However, these methods often assume that the similarity between a trajectory and an expert demonstration is positively correlated with the reward, which oversimplifies the underlying reward structure. We propose PROF, a novel framework that leverages large language models (LLMs) to generate and improve executable reward function codes from natural language descriptions and a single expert trajectory. We propose Reward Preference Ranking (RPR), a novel reward quality assessment and ranking strategy without requiring environment interactions or RL training. RPR calculates the dominance scores of the reward functions, where higher scores indicate better alignment with expert preferences. By alternating between RPR and text-based gradient optimization, PROF fully automates the selection and refinement of optimal reward functions for downstream policy learning. Empirical results on D4RL demonstrate that PROF surpasses or matches recent strong baselines across numerous datasets and domains in D4RL, highlighting the effectiveness of our approach.

## 1 INTRODUCTION

Reinforcement learning (RL) (Kaelbling et al., 1996) has achieved remarkable successes across diverse domains such as games (Mnih et al., 2013; 2015; Silver et al., 2016; OpenAI et al., 2019) and robotics (Yu et al., 2020; Gu et al., 2023; Rajeswaran et al., 2017). Offline RL (Lange et al., 2012; Levine et al., 2020) extends this success by enabling the learning of decision-making policies directly from previously collected data, without requiring further interaction with the environment. Its effectiveness has been consistently demonstrated in prior studies (Fujimoto & Gu, 2021; Kostrikov et al., 2022; Li et al., 2024a;c; Lyu et al., 2025; 2022b;a; Tarasov et al., 2024). However, offline RL typically requires reward signals for each transition, which are often unavailable in practical settings. Designing reward functions manually (Laud, 2004; Gupta et al., 2022) is not only time-consuming and dependent on domain expertise, but it can also lead to suboptimal (Booth et al., 2023) or unintended behaviors (Hadfield-Menell et al., 2017). As an alternative, offline imitation learning (offline IL) addresses this challenge by behavior cloning (BC) (Pomerleau, 1988) or offline inverse reinforcement learning (offline IRL) (Kostrikov et al., 2020; Kim et al., 2022b).

Recent works (Zolna et al., 2020; Yu et al., 2022; Luo et al., 2023; Lyu et al., 2024) have explored leveraging expert demonstrations to annotate rewards for offline datasets by comparing them with unlabeled trajectories. These methods decouple reward labeling from RL training and achieve promising results using only a limited number of expert examples. However, they typically rely on distance metrics between trajectories to infer rewards, which biases learning toward trajectories that closely resemble expert behavior. This overlooks the possibility that optimal behaviors may be diverse and not necessarily proximal to a limited set of demonstrations. Moreover, the reward signals generated in this manner are often not easily interpretable or adjustable by humans, limiting their utility in safety-critical applications. Alternatively, recent efforts (Yu et al., 2023; Xie et al., 2024; Ma et al., 2024; Sun et al., 2025; Qu et al., 2025) leverage the semantic understanding capabilities of large language models (LLMs) to generate executable reward function codes. While promising,

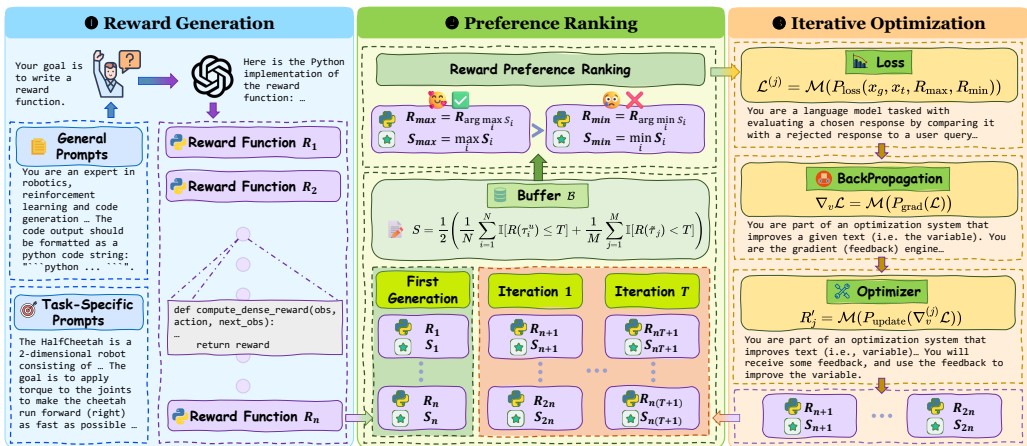

Figure 1: **The framework of PROF**. PROF initiates by generating $n$ candidate reward functions, which are stored in a buffer $\mathcal{B}$. The algorithm proceeds through $T$ rounds of iterative optimization. In each round, the reward functions with the highest and lowest dominance scores are selected from the buffer to construct the loss feedback. Leveraging TextGrad, gradients are computed automatically and backpropagation is applied to optimize each candidate independently, yielding $n$ new reward functions. These newly optimized candidates are added to the buffer, ensuring diversity and continual improvement. After $T$ iterations, PROF outputs the reward function with the highest dominance score.

these methods are primarily designed for online settings, as they depend on continual interaction with the environment to refine the reward codes iteratively.

In this paper, we introduce PROF, an LLM-based Reward Code **PR**eference **O**ptimization **F**ramework for offline IL. PROF leverages LLMs to generate reward function codes, which are subsequently refined through preference optimization. To guide this process, we introduce two fundamental principles for evaluating any reward function: first, the return of expert demonstrations should be larger than that of any trajectory contained in the offline dataset; second, expert demonstrations must be rewarded higher than their corresponding noisy variants. Leveraging these principles, we propose Reward Preference Ranking (RPR), which enables efficient reward quality assessment and preference ranking using only one expert trajectory, without requiring environment interactions and RL training. As illustrated in Figure 1, PROF begins by providing environmental information to the LLM, prompting it to generate initial reward function candidates. These candidates are then evaluated and ranked based on predefined criteria to establish relative preferences. To refine the codes, PROF applies a text gradient technique (Yuksekgonul et al., 2025), which optimizes code quality by exploiting preference relationships. Through repeated cycles of ranking and optimization, the reward function codes improve progressively. After a fixed number of iterations, the final optimized code is employed to annotate rewards for the offline dataset, enabling downstream offline RL. Empirical results on D4RL demonstrate that PROF consistently outperforms recent strong reward labeling baselines across a diverse set of domains. Remarkably, by leveraging only a single expert trajectory, PROF enables offline RL algorithms to match or even surpass the oracle.

## 2 RELATED WORK

**Offline Imitation Learning.** Offline imitation learning (offline IL) differs from offline RL in that it does not assume access to reward signals. Behavior Cloning (BC) is the most straightforward approach for offline RL, directly applying supervised learning to mimic expert behavior (Pomerleau, 1988). However, BC suffers from compounding errors (Rajaraman et al., 2020). To address this challenge, offline inverse reinforcement learning (offline IRL) either optimizes policies under additional constraints (Jarrett et al., 2020; Xu et al., 2022; Dadashi et al., 2021; Kostrikov et al., 2020) or recovers a reward function from offline data followed by policy optimization (Chang et al., 2021; Kim et al., 2022a;b; Ma et al., 2022; Yue et al., 2023). An alternative research direction

annotates offline datasets with rewards using auxiliary utilities (Reddy et al., 2020; Zolna et al., 2020; Yu et al., 2022; Luo et al., 2023; Lyu et al., 2024), effectively converting offline IL into an offline RL problem. For example, ORIL (Zolna et al., 2020) infers rewards by contrasting expert demonstrations with unlabeled trajectories. UDS (Yu et al., 2022) simply assigns minimal rewards to unlabeled data while preserving expert data. OTR (Luo et al., 2023) utilizes optimal transport to assign rewards with state-action pairs, and SEABO (Lyu et al., 2024) employs a KD-tree structure to generate dense reward signals. Distinct from these prior efforts, our method leverages LLMs to automatically generate executable reward function code, offering a promising framework for reward design in offline IL.

**Reward Design via Large Language Models.** Recent advances in LLMs (OpenAI, 2023; Hurst et al., 2024; Anthropic, 2023) have sparked increasing interest in utilizing them to facilitate RL training. Existing approaches can be broadly categorized into three lines of research. The first prompts LLMs to act as proxy reward functions, guiding RL agents through extensive query interactions (Ma et al., 2023a;b; Fan et al., 2022; Kwon et al., 2023; Du et al., 2023). The second directly generates policy code using LLMs (Liang et al., 2022; Silver et al., 2024; Deng et al., 2024). The third focuses on instructing LLMs to produce executable reward function codes for policy learning (Yu et al., 2023; Li et al., 2024b; Zeng et al., 2024b; Qu et al., 2025). Within the third line, various techniques have been proposed to improve reward quality. Text2Reward (Xie et al., 2024) and ICPL (Yu et al., 2024) incorporate human feedback to refine the rewards. Auto-MC-Reward (Li et al., 2024b) leverages LLMs to analyze trajectories and generate feedback. Eureka (Ma et al., 2024) and CARD (Sun et al., 2025) construct feedback from RL training outcomes, while Video2Reward (Zeng et al., 2024a) employs video-assisted schemes for reward refinement. However, these methods mainly focus on reward design for online RL. In contrast, our approach targets reward function generation in the offline RL setting, where direct interaction with the environment is not feasible.

# 3 PRELIMINARIES

**Reinforcement Learning.** We consider the standard Markov Decision Process (MDP) (Sutton & Barto, 2018) represented by a tuple $(\mathcal{S}, \mathcal{A}, P, \mathcal{R}, \gamma)$, where $\mathcal{S}$ is the state space, $\mathcal{A}$ is the action space, $P$ is the transition dynamics, $\mathcal{R} : \mathcal{S} \times \mathcal{A} \times \mathcal{S} \to \mathbb{R}$ is the reward function, such that $r_t = \mathcal{R}(s_t, a_t, s_{t+1})$, and $\gamma \in [0, 1]$ is the discount factor. In the offline IL setting, we cannot access the reward function. Instead, we have expert demonstrations $\mathcal{D}_e = \{\tau_e^i\}_{i=1}^K$ and an offline dataset $\mathcal{D}_u = \{\tau_u^i\}_{i=1}^N$ from unknown behavior policies. The goal of offline IL is to learn a well-behaved policy from both the expert demonstrations and the unlabeled dataset $\mathcal{D} = \mathcal{D}_e \cup \mathcal{D}_u$.

**Text Gradient.** Recent work (Li et al., 2025; Yuksekgonul et al., 2025) introduces TextGrad, a novel approach that differs from traditional gradient-based optimization (Rafailov et al., 2023; Shao et al., 2024) by optimizing the model output through searching for the optimal context. Notably, this method computes gradients entirely in textual form, enabling direct optimization of text variables, e.g., the output of LLMs, without requiring any fine-tuning of the model parameters.

Let $\mathcal{M}$ denote a frozen LLM, and $P$ a prompt function incorporating instructions or preferences. Given a query $x$, we define the model output $v \leftarrow \mathcal{M}(x)$ as the optimization variable. The process begins with the prompt $P_{\text{loss}}$, which elicits from the model $\mathcal{M}$ a preference judgment over candidate outputs, effectively identifying desirable and undesirable aspects of the text. Building upon this, the model is guided to perform an introspective analysis, determining the reasons behind the varying preferences.

$$\mathcal{L}(x, v) \coloneqq \mathcal{M}\big(P_{\text{loss}}(x, v)\big). \tag{1}$$

Based on this analysis, the prompt $P_{\text{grad}}$ instructs the model to compute a textual gradient that captures directional suggestions for improving $v$,

$$\nabla_v \mathcal{L}(x, v) \coloneqq \mathcal{M}\big(P_{\text{grad}}(\mathcal{L}(x, v))\big). \tag{2}$$

Finally, $P_{\text{update}}$ prompts the model to revise the text according to the computed gradient, in a manner analogous to the parameter update rule $\theta \leftarrow \theta - \alpha \nabla_\theta \mathcal{L}(\theta)$.

$$v_{\text{new}} \coloneqq \mathcal{M}\big(P_{\text{update}}(\nabla_v \mathcal{L}(x, v))\big). \tag{3}$$

## 4 METHOD

In this section, we present PROF, which involves three key steps: (i) **Reward Generation** (▷ Section 4.1): PROF prompts the LLM to generate a diverse set of reward function candidates conditioned on environmental descriptions; (ii) **Preference Ranking** (▷ Section 4.2): These candidates are then evaluated and ranked using the Reward Preference Ranking (RPR); and (iii) **Iterative Optimization** (▷ Section 4.3): The top-ranked reward function is refined and optimized through code-level adjustments, guided by the TextGrad (Yuksekgonul et al., 2025), forming an automatic optimization cycle.

### 4.1 REWARD GENERATION

Following prior works (Xie et al., 2024; Ma et al., 2024; Sun et al., 2025), we query LLMs in a zero-shot setting to generate executable reward function codes in Python, providing only task-related prior knowledge through designed prompts. This approach enables inherent generalization and avoids domain-specific fine-tuning. Our prompt design is structured into two components: general prompts and task-specific prompts.

General prompts provide a consistent foundation across tasks by defining the expert role of LLMs, clarifying reward design, supplying a reward function template, specifying coding standards and constraints, guiding the thinking process, and offering instructions for designing reward functions. These elements remain constant across environments. Task-specific prompts complement them with details of a particular RL task, including its objective and the complete definition of the observation and action spaces.

Given the above prompts, PROF queries LLMs to generate reward functions at this stage. However, codes produced by LLMs often contain syntax or runtime errors, such as undefined variables or incorrect matrix dimensions. Inspired by Ma et al., we generate multiple independent reward function candidates in parallel to ensure that at least one is executable. Details of the prompts are shown in Appendix C.

### 4.2 PREFERENCE RANKING

Prior works (Xie et al., 2024; Li et al., 2024b; Ma et al., 2024; Sun et al., 2025) show that LLMs often struggle to generate high-quality reward functions in a single attempt. To address this, methods typically sample various responses in parallel for broader coverage and iteratively refine them based on feedback. Feedback may come from humans (Xie et al., 2024), LLMs (Li et al., 2024b), or automated sources (Ma et al., 2024; Sun et al., 2025) within online RL. These approaches generally rely on task success rates in online evaluations to select optimal reward functions across iterations and query batches, while also exploiting RL training results to construct feedback for further refinement. However, such methods are not applicable in offline IL, where environment access is restricted. Moreover, expert data is typically scarce due to high collection costs. This necessitates an algorithm capable of evaluating reward functions and constructing feedback without environment interaction, using only a small number of expert demonstrations and a large amount of unlabeled offline data of unknown quality. To address these challenges, we propose Reward Preference Ranking, a preference-based reward function evaluation and ranking algorithm tailored for offline IL.

Our work is motivated by two fundamental insights. First, the return of the expert demonstration should be at least as high as the return of any trajectory in the offline dataset. Second, the return of an expert trajectory should exceed that of a perturbed version with random noise. These insights reflect the intuition that expert behavior ought to be not only superior to suboptimal offline data but also robust to noise, providing a stable reference for reward design. Since the optimal trajectory may not be unique and other expert trajectories might also be in the dataset, we allow a small margin of tolerance.

To formalize this, Reward Preference Ranking computes a score based on the proportion of expert trajectories that outperform offline dataset trajectories and noise-perturbed trajectories, measuring the superiority of expert demonstrations. A higher score indicates a clearer superiority of expert behavior, suggesting that the reward function more accurately captures the intended task and aligns with human preferences. The complete algorithm is formally defined as follows.

**Expert Demonstration Return Threshold.** We assume the access to a static offline dataset $\mathcal{D}_u = \{\tau_i^u\}_{i=1}^N$ along with a limited set of expert demonstrations $\mathcal{D}_e = \{\tau_j^e\}_{j=1}^K$. Each trajectory $\tau$ is a sequence of transitions $\{(o_t, a_t, r_t, o_{t+1})\}_{t=1}^{|\tau|-1}$, with return $R(\tau) = \sum_{t=1}^{|\tau|-1} r_t$. To quantify expert trajectories, we introduce a return threshold $\lambda$ with a tolerance parameter $\delta \in [0, 1]$ to enable consistent comparison across different trajectories.

$$\lambda = \begin{cases} (1 + \delta) \cdot \min_j R(\tau_j^e), & \text{if } \min_j R(\tau_j^e) \geq 0, \\ (1 - \delta) \cdot \min_j R(\tau_j^e), & \text{otherwise.} \end{cases} \tag{4}$$

**Noisy Trajectory Construction.** To simulate suboptimal or disturbed behaviors, we generate noisy variants of expert trajectories by injecting Gaussian noise into the observations and actions of each transition. The noise is applied relative to the expert trajectory $\tau_{\min}^e = \arg\min_j R(\tau_j^e)$. To account for the varying scales of observation and action spaces across different domains, we define adaptive noise scales $\sigma_o$ and $\sigma_a$ based on the trajectory-level standard deviations of observations and actions in $\tau_{\min}^e$:

$$\sigma_o = \alpha_o \cdot \text{std}\left(\{o_t\}_{t=1}^{|\tau|-1}\right), \qquad \sigma_a = \alpha_a \cdot \text{std}\left(\{a_t\}_{t=1}^{|\tau|-1}\right), \tag{5}$$

where $\alpha_o > 0$ and $\alpha_a > 0$ are predefined scaling hyperparameters.

We now define the process of generating noisy trajectories from $\tau_{\min}^e = \{(o_t, a_t, r_t, o_{t+1})\}_{t=1}^{|\tau_{\min}^e|-1}$. Let $H$ be the number of noisy trajectories to generate. Let $\mathcal{D}_n = \{\tau_h^n\}_{h=1}^H$ be the noisy trajectory set. Let $\mathcal{N}(\mu, \sigma^2)$ be the Gaussian distribution, where $\mu$ and $\sigma$ are the mean and standard deviation. For each $h \in \{1, \ldots, H\}$, construct

$$\widetilde{\tau}_m = \{(\tilde{o}_t, \tilde{a}_t, r_t, \tilde{o}_{t+1})\}_{t=1}^{|\tau_{\min}^e|-1}, \tag{6}$$

where

$$\tilde{o}_t = o_t + \mathcal{N}(0, (\sigma_o)^2), \qquad \tilde{a}_t = a_t + \mathcal{N}(0, (\sigma_a)^2), \qquad \tilde{o}_{t+1} = o_{t+1} + \mathcal{N}(0, (\sigma_o)^2), \tag{7}$$

for $t = 1, \ldots, T - 1$. Note that the last transition of the trajectory $(o_{T-1}, a_{T-1}, r_{T-1}, o_T)$ is excluded from noise construction. This transition often contains critical information that determines whether the goal has been successfully achieved. Introducing noise at this stage can distort outcome labels, potentially converting a failed trajectory into a successful one or a successful trajectory into a failure, thereby compromising the reliability of the learning signal.

**Dominance Score.** Finally, we define the dominance score, which quantifies the relative superiority of expert demonstrations over both the offline dataset and its noisy perturbations.

$$S = \frac{1}{2}\left(\frac{1}{N}\sum_{i=1}^N \mathbb{I}\left[R(\tau_i^u) \leq \lambda\right] + \frac{1}{H}\sum_{j=1}^H \mathbb{I}\left[R(\widetilde{\tau}_j) < \lambda\right]\right). \tag{8}$$

**Reward Preference Ranking.** For a set of reward functions $\{R_1, R_2, \ldots, R_n\}$, RPR returns the reward function with the highest dominance score as the result, since a higher score reflects better task understanding and closer alignment with human preferences.

## 4.3 ITERATIVE OPTIMIZATION

Prior works (Xie et al., 2024; Ma et al., 2024; Li et al., 2024b; Sun et al., 2025) have shown that iterative improvement is often necessary to further optimize reward quality. Inspired by recent studies (Li et al., 2025; Sun et al., 2025), we utilize the preference relationships derived from RPR between reward functions to guide reward optimization. Instead of treating the dominance scores of reward functions as absolute metrics, we interpret them through the lens of comparative preference. This paradigm offers a more stable and robust signal for optimization, especially when absolute scores are noisy or poorly calibrated. Specifically, we follow the autograd engine introduced by Yuksekgonul et al. to perform iterative optimization and keep all prompts related to it unchanged. Formally, given $n$ sampled reward functions $R_i \leftarrow \mathcal{M}(x), i = 1, \ldots, n$, general prompts $x_g$ and

task-specific prompts $x_t$ as described in Section 4.1, we use Reward Preference Ranking function $F_r$ to compute their scores:

$$S_i = F_r(R_i), \quad i = 1, 2, \ldots, n. \tag{9}$$

We then update the optimal reward function code $R_{\arg\max_i S_i}$. First, we define a textual loss as the difference between the reward functions with the highest and lowest dominance scores, based on equation 1. TextGrad then computes this loss by prompting the LLM to reflect on the rationale behind the superiority of the higher-scoring reward function:

$$\mathcal{L} = \mathcal{M}\big(P_{\text{loss}}(x_g, x_t, R_{\arg\max_i S_i}, R_{\arg\min_i S_i})\big). \tag{10}$$

Similar to equation 2, TextGrad calculates the gradient of the loss by generating actionable suggestions for improvement:

$$\nabla_v \mathcal{L} = \mathcal{M}\big(P_{\text{grad}}(\mathcal{L})\big). \tag{11}$$

These suggestions guide an automatic optimization step according to equation 3, resulting in an updated, more effective reward function:

$$R' = \mathcal{M}\big(P_{\text{update}}(\nabla_v \mathcal{L})\big), \tag{12}$$

where $R'$ denotes the reward function obtained after optimization based on $R_{\arg\max_i S_i}$. PROF maintains a dynamic buffer $\mathcal{B}$ of reward functions, populated by $n$ initial functions derived solely from the prompts $x_g$ and $x_t$. In each subsequent iteration, the framework identifies the highest and lowest scoring reward functions within the buffer to construct the loss feedback, guiding the independent optimization of $n$ new reward functions. These optimized functions are then added to the buffer, and the process repeats until a predefined number of iterations $T$ is reached. We employ the highest-scoring reward function obtained from the entire buffer after iteration termination to label offline dataset rewards. Prompt templates used in the feedback construction are in Appendix C.3. The pseudo code of PROF is presented in Appendix B.

## 5 EXPERIMENTS

We conduct experiments to answer the following questions: (**Q1**) How well does PROF perform given only one expert demonstration compared to the baselines? (**Q2**) How sensitive is PROF to its key parameters? (**Q3**) Does PROF improve various offline RL algorithms? Additionally, we discuss further questions in Appendix E: (**Q4**) How does PROF perform compared to imitation learning algorithms? (**Q5**) How does PROF perform when demonstrations contain only observations?

### 5.1 SETUP

We conduct experiments on the widely adopted D4RL (Fu et al., 2020) benchmark, discarding the original reward signals to construct an unlabeled dataset. Following prior works (Luo et al., 2023; Lyu et al., 2024), we use the trajectory with the highest return in the original dataset as the expert trajectory. All experiments are conducted consistently utilizing only one single expert trajectory ($K = 1$). PROF adopts a zero-shot setting and utilizes the `GPT-4o-2024-11-20` (Hurst et al., 2024) API unless otherwise specified. For all tasks, the tolerance parameter $\delta$ is fixed at $0.01$, the scaling parameters $\alpha_o$ and $\alpha_a$ are set to $0.05$, and the number of noisy trajectories is $H = 10^4$. Our method runs all experiments once in full. For MuJoCo environments, we perform 1 round of reward generation followed by $T = 1$ rounds of iterative optimization, while in AntMaze and Adroit we conduct 1 reward generation round and $T = 2$ optimization rounds. Each round involves $n = 5$ independent samplings. Each experiment is run for 1 million gradient steps using 5 distinct random seeds, and we report the mean D4RL normalized score at the final step along with the standard deviation. All results of baselines are sourced directly from the SEABO paper (Lyu et al., 2024). Complete experimental details and hyperparameter configurations are provided in Appendix D.

**Baselines.** We compare PROF against the following baselines: (i) **BC** (Pomerleau, 1988) that mimics expert behavior using supervised learning. (ii) **IQL** (Kostrikov et al., 2022) that learns from offline datasets using ground-truth rewards without querying out-of-distribution (OOD) actions. (iii) **ORIL** (Zolna et al., 2020) that contrasts expert demonstrations with unlabeled trajectories to infer rewards. (iv) **UDS** (Yu et al., 2022), which retains rewards from expert data while assigning minimal

Table 1: **Comparison of PROF and baselines on D4RL MuJoCo locomotion tasks.** We report the mean D4RL normalized score with standard deviation, calculated across 5 random seeds. We bold and shade the cells with the highest scores in each task. Abbreviations: hc = halfcheetah, hop = hopper, w2d = walker2d; m = medium, mr = medium-replay, me = medium-expert.

| Task | BC | IQL | IQL+ORIL | IQL+UDS | IQL+OTR | IQL+SEABO | IQL+PROF |
|------|------|------|----------|---------|---------|-----------|----------|
| hc-m | 42.6 | 47.4±0.2 | **49.0±0.2** | 42.4±0.3 | 43.2±0.2 | 44.8±0.3 | 47.4±0.1 |
| hop-m | 52.9 | 66.2±5.7 | 47.0±4.0 | 54.5±3.0 | 74.2±5.1 | **80.9±3.2** | 65.9±3.8 |
| w2d-m | 75.3 | 78.3±8.7 | 61.9±6.6 | 68.9±6.2 | 78.7±2.2 | 80.9±0.6 | **82.2±1.2** |
| hc-mr | 36.6 | **44.2±1.2** | 44.1±0.6 | 37.9±2.4 | 41.8±0.3 | 42.3±0.1 | 42.6±2.2 |
| hop-mr | 18.1 | 94.7±8.6 | 82.4±1.7 | 49.3±22.7 | 85.4±0.8 | 92.7±2.9 | **96.6±3.1** |
| w2d-mr | 26.0 | 73.8±7.1 | 76.3±4.9 | 17.7±9.6 | 67.2±6.0 | 74.0±2.7 | **82.4±1.8** |
| hc-me | 55.2 | 86.7±5.3 | 87.5±3.9 | 63.0±5.7 | 87.4±4.4 | 89.3±2.5 | **89.6±3.2** |
| hop-me | 52.5 | 91.5±14.3 | 29.7±22.2 | 53.9±2.5 | 88.4±12.6 | 97.5±5.8 | **108.3±5.1** |
| w2d-me | 107.5 | 109.6±1.0 | 110.6±0.6 | 107.5±1.7 | 109.5±0.3 | **110.9±0.2** | 109.8±0.7 |
| total | 466.7 | 692.4 | 588.5 | 495.1 | 675.8 | 713.3 | **724.8** |

rewards to unlabeled data. (v) **OTR** (Luo et al., 2023) that employs optimal transport to align expert and unlabeled data for reward inference. (vi) **SEABO** (Lyu et al., 2024), which leverages a KD-tree to identify the nearest expert transition and assigns rewards based on the distance between unlabeled and expert transitions. All methods except BC adopt IQL as the base RL algorithm.

## 5.2 COMPARISON OF PROF WITH BASELINES (Q1)

**Experiments on Locomotion Tasks.** We compare PROF with baselines on three D4RL MuJoCo locomotion tasks: *Half Cheetah*, *Hopper*, and *Walker2d*. For each task, we utilize 3 medium-level datasets: *medium-v2*, *medium-replay-v2*, and *medium-expert-v2*, resulting in a total of 9 task-dataset combinations. Note that the original OTR computes rewards based solely on observations, whereas both SEABO and PROF leverage $(s, a, s')$ tuples for reward design. To enable a fair comparison, we report the results of modified OTR from the SEABO paper. This version of OTR also adopts $(s, a, s')$ to compute rewards on MuJoCo tasks. The corresponding results are presented in Table 1. We observe that: **Obs.❶ PROF significantly outperforms all baselines on locomotion tasks.** Unlike approaches that construct rewards solely based on proximity to one single expert demonstration, PROF leverages human-aligned reward design, which better captures the true distribution of rewards when optimal trajectories are not unique. PROF achieves the best performance on 5 out of 9 tasks, demonstrating its effectiveness. On other tasks, it remains competitive, except for *hopper-medium*, where it performs notably worse than OTR and SEABO. We attribute this to the limited diversity of successful trajectories in this medium-quality dataset, where mimicking expert behavior is more effective. Overall, PROF achieves the highest total score across all tasks. In addition to its strong empirical performance, it offers interpretable, human-readable reward functions and enables further improvement through expert feedback.

**Experiments on Challenging Tasks.** We further evaluate PROF on the AntMaze from D4RL, using 6 "v0" datasets: *umaze*, *umaze-diverse*, *medium-diverse*, *medium-play*, *large-diverse*, and *large-play*. Results in Table 2 demonstrate that: **Obs.❷ PROF also significantly outperforms baselines on complex goal-conditioned tasks.** Specifically, PROF surpasses the strong baselines on 5 out of 6 tasks and achieves the highest total score. To better illustrate the advantages of PROF, we also report the improvement percentage of each algorithm relative to IQL trained with ground-truth rewards. Notably, PROF exceeds all baselines in terms of average improvement ratio, demonstrating its superiority across various tasks. We also evaluate the performance of PROF on the challenging Adroit domain as presented in Appendix E.1. We observe that: **Obs.❸ PROF substantially enhances the improvement over IQL on manipulation tasks.** PROF achieves the largest improvement over IQL compared to other baselines in this domain. When combined with PROF, IQL achieves a 102.3% increase in performance. In contrast, SEABO improves IQL by only 52.2%, while OTR fails to fully recover the performance of IQL trained with ground-truth rewards and results in a 5.7% degradation. These findings demonstrate the ability of PROF to accurately model complex reward distributions and enhance policy learning beyond what is achievable with ground-

Table 2: **Comparison of PROF and baselines on D4RL AntMaze tasks.** We report the mean D4RL normalized score with standard deviation, calculated across 5 random seeds. For each task, the value in parentheses denotes the relative improvement of the "IQL+" algorithms compared to the IQL trained with ground-truth rewards. For the "total" row, the value in parentheses indicates the average relative improvement across all tasks. We bold and shade the cells with the highest scores in each task.

| Task | IQL | IQL+OTR | IQL+SEABO | IQL+PROF |
|---|---|---|---|---|
| umaze | 87.5±2.6 | 83.4±3.3 (-4.7%) | 90.0±1.8 (+2.9%) | **93.0±3.9 (+6.3%)** |
| umaze-diverse | 62.2±13.8 | 68.9±13.6 (+10.8%) | 66.2±7.2 (+6.4%) | **69.0±9.1 (+10.9%)** |
| medium-diverse | 70.0±10.9 | 70.4±4.8 (+0.6%) | 72.2±4.1 (+3.1%) | **75.8±5.8 (+8.3%)** |
| medium-play | 71.2±7.3 | 70.5±6.6 (-1.0%) | 71.6±5.4 (+0.6%) | **76.6±3.3 (+7.6%)** |
| large-diverse | 47.5±9.5 | 45.5±6.2 (-4.2%) | 50.0±6.8 (+5.3%) | **51.6±4.5 (+8.6%)** |
| large-play | 39.6±5.8 | 45.3±6.9 (+14.4%) | **50.8±8.7 (+28.3%)** | 43.4±10.9 (+9.6%) |
| total | 378.0 | 384.0 (+2.7%) | 400.8 (+7.8%) | **409.4 (+8.6%)** |

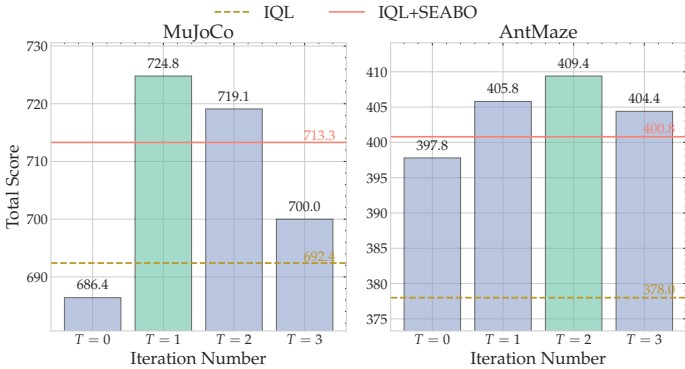

Figure 2: **The performance of PROF across different numbers of iterative optimization rounds.** From left to right are MuJoCo and AntMaze domains. We report the total D4RL normalized score calculated across 5 seeds for each domain.

truth reward signals. We present a case study in Appendix F that explains why the reward functions designed by PROF significantly outperform the ground-truth reward functions.

## 5.3 ABLATION STUDY (Q2)

**Different Iterative Optimization Numbers.** We evaluate PROF on the D4RL benchmark tasks across MuJoCo and AntMaze domains with various numbers of iterations $T$. All prompt settings and parameters follow Section 5.1. $T = i$ indicates that the reward function used for RL training is the one with the highest dominance score in buffer $\mathcal{B}$ after the $i$-th iteration of optimization. $T = 0$ means no iterative optimization is performed. Notably, the highest-scoring reward function may remain unchanged across iterations. Results in Figure 2 reveal a non-monotonic relationship between $T$ and performance. We offer the following key observation: **Obs.❹ As $T$ increases, performance initially improves before declining**. For relatively simple environments like MuJoCo, the best performance is observed at $T = 1$, although $T = 2$ still surpasses the previous strong method SEABO. In contrast, more complex environments such as AntMaze achieve optimal results at $T = 2$. This suggests that moderate iteration enables LLMs to design higher-quality reward functions, but further iterations saturate due to limited information and risk introducing instability through reward hacking. Experiments on the Adroit domain confirm this trend, with $T = 2$ achieving the best results, as shown in Appendix E.1. These results provide valuable guidance for reward design in offline scenarios, suggesting that $T = 1$ is appropriate for simple tasks while $T = 2$ is preferable for more complex tasks. Notably, similar conclusions have also been reported in prior work (Sun et al., 2025). Detailed results for each task across iterations are provided in Appendix E.2. A code-level analysis detailing the changes throughout the iteration process is provided in Appendix G.

Table 3: **Comparison of baselines and PROF using different LLM APIs on D4RL *Half Cheetah* tasks.** We report the mean D4RL normalized score with standard deviation, calculated across 5 random seeds. We bold and shade the cells if scores of PROF match or surpass the previous strong method SEABO. Abbreviations: hc = halfcheetah, hop = hopper, w2d = walker2d; m = medium, mr = medium-replay, me = medium-expert.

| Task | IQL | IQL+SEABO | PROF (GPT-4o) | PROF (DeepSeek V3) | PROF (Claude 3.7 Sonnet) |
|---|---|---|---|---|---|
| hc-m | 47.4±0.2 | 44.8±0.3 | **47.4±0.1** | **44.8±0.1** | **45.9±0.1** |
| hc-mr | 44.2±1.2 | 42.3±0.1 | **42.6±2.2** | **45.3±0.6** | **43.0±1.5** |
| hc-me | 86.7±5.3 | 89.3±2.5 | **89.6±3.2** | **90.6±2.9** | 88.9±3.0 |
| total | 178.3 | 176.4 | **179.6** | **180.7** | **177.8** |

Table 4: **Comparison of baselines and PROF on D4RL *Half Cheetah* tasks with TD3_BC as the base algorithm.** We report the mean normalized score with standard deviation, calculated across 5 seeds. $\mu_{max}$ denotes the normalized return of the highest return trajectory in the specific dataset. We bold and shade the cells if scores of PROF match or surpass SEABO. Abbreviations: hc = halfcheetah, hop = hopper, w2d = walker2d; m = medium, mr = medium-replay, me = medium-expert.

| Task | $\mu_{max}$ | TD3_BC | TD3_BC+ SEABO | TD3_BC+ PROF | IQL | IQL+ SEABO | IQL+ PROF |
|---|---|---|---|---|---|---|---|
| hc-m | 45.0 | 48.0±0.7 | 45.9±0.3 | **54.3±0.7** | 47.4±0.2 | 44.8±0.3 | **47.4±0.1** |
| hc-mr | 42.4 | 44.4±0.8 | 43.0±0.4 | **46.2±0.1** | 44.2±1.2 | 42.3±0.1 | **42.6±2.2** |
| hc-me | 92.8 | 93.5±2.0 | 95.7±0.4 | 94.8±1.1 | 86.7±5.3 | 89.3±2.5 | **89.6±3.2** |
| total | 180.2 | 185.9 | 184.6 | **195.3** | 178.3 | 176.4 | **179.6** |

**Different LLM APIs.** To assess the generalizability of PROF across diverse LLM APIs, we extend our evaluation to two widely used APIs: `DeepSeek-V3-0324` (Liu et al., 2024) and `Claude 3.7 Sonnet`[1]. Experiments are conducted on three *Half Cheetah* "v2" datasets (*medium*, *medium-replay*, and *medium-expert*), following the same experimental setup described in Section 5.1. As presented in Table 3, we observe that: **Obs.⑤ PROF consistently surpasses the strong baseline SEABO across diverse LLM APIs**, demonstrating its robustness and adaptability.

### 5.4 EXPERIMENTS ON VARIOUS OFFLINE RL ALGORITHMS (Q3)

We investigate the applicability of PROF across diverse offline RL algorithms. To this end, we conduct experiments on 3 *Half Cheetah* "v2" datasets (*medium*, *medium-replay*, and *medium-expert*) using two widely adopted offline RL methods: TD3_BC (Fujimoto & Gu, 2021) and IQL (Kostrikov et al., 2022). Results in Table 4 indicate that: **Obs.⑥ PROF effectively enhances multiple offline RL algorithms.** Specifically, integrating PROF with TD3_BC leads to a substantial enhancement in total score. Furthermore, when PROF is combined with IQL, it consistently surpasses SEABO across all three tasks.

### 6 CONCLUSION

We propose PROF, a fully automatic framework for reward function generation and optimization in offline IL. PROF integrates the generative capabilities of LLMs, a novel preference ranking algorithm (Reward Preference Ranking) based on dominance scores, and a textual optimization method (TextGrad). This combination enables reward design without interacting with the environment or performing RL training. By generating human-interpretable reward function code and effectively capturing the underlying reward distribution, PROF achieves similar or better performance against strong baselines on the D4RL benchmark. These results highlight the potential of PROF as a practical reward design method in offline IL settings.

---

[1]https://www.anthropic.com/news/claude-3-7-sonnet

## 7 ETHICS STATEMENT

To the best of our knowledge, this work does not present any ethical concerns.

## 8 REPRODUCIBILITY STATEMENT

The source code provided in the supplementary materials ensures that the algorithm and experimental results reported in this paper can be fully reproduced.

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

## A  THE USE OF LARGE LANGUAGE MODELS

LLMs are used via APIs for designing and refining the reward functions in our algorithm. For manuscript preparation, LLMs are used only for grammar checking and polishing. They are not involved in retrieval or ideation.

## B  ALGORITHM

The pseudo code of PROF is presented in Algorithm 1.

---

**Algorithm 1**: An LLM-Based Reward Code PReference Optimization Framework for Offline IL (PROF)

---

**Input**: Offline dataset $\mathcal{D}_u = \{\tau_i^u\}_{i=1}^N$, expert demos $\mathcal{D}_e = \{\tau_j^e\}_{j=1}^K$, general prompt $x_g$, task-specific prompt $x_t$, number of candidates $n$, max iterations $T$, number of noisy samples $H$, noise scales $\alpha_o, \alpha_a$

**Output**: Optimal reward function $R^*$

1: **// Generate noisy trajectories**
2: Compute return threshold $\lambda$ using equation 4
3: Identify $\tau_{\min}^e = \arg\min_j R(\tau_j^e)$
4: Compute noise scales $\sigma_o, \sigma_a$ using equation 5
5: Generate noisy dataset $\mathcal{D}_n$ by sampling $H$ trajectories via equation 6 and equation 7
6: **// Code generation**
7: **for** $i = 1$ to $n$ **do**
8:     Sample reward $R_i \leftarrow \mathcal{M}(x_g, x_t)$
9:     Compute score $S_i = F_r(R_i)$ using equation 8 and equation 9
10:    Add pair $(R_i, S_i)$ to buffer $\mathcal{B}$
11: **end for**
12: **for** $t = 1$ to $T$ **do**
13:     **// Preference Ranking**
14:     Identify $R_{\max} = R_{\arg\max_i F_r(R_i)}$, $R_{\min} = R_{\arg\min_i F_r(R_i)}$
15:     **// Iterative Optimization**
16:     Initialize empty set of new rewards $\mathcal{B}' \leftarrow \emptyset$
17:     **for** $j = 1$ to $n$ **do**
18:         Calculate loss $\mathcal{L}$ using equation 10
19:         Calculate gradient $\nabla_v \mathcal{L}$ using equation 11
20:         Obtain an optimized function $R_j'$ using equation 12
21:         Compute score $S_j' = F_r(R_j')$ using equation 8 and equation 9
22:         Add pair $(R_j', S_j')$ to $\mathcal{B}'$
23:     **end for**
24:     Update buffer: $\mathcal{B} \leftarrow \mathcal{B} \cup \mathcal{B}'$
25: **end for**
26: **return** $R^* = R_{\arg\max_{(R,S)\in\mathcal{B}} S}$

---

## C  PROMPT DETAILS

### C.1  GENERAL PROMPTS

The general prompts are detailed in Listing 1. We employ an identical general prompt across all experiments, demonstrating the generalization capabilities of our approach. In ablation studies where only $(s, s')$ is provided as input, we adapt the reward function template by replacing the standard $(s, a, s')$ input with $(s, s')$, while preserving all other components. Notably, the design of the general prompt, based on previous works Xie et al. (2024); Ma et al. (2024); Sun et al. (2025); Qu et al. (2025), is simple, easy to create, and consistent across all experiments, showing that it can be applied to a wide range of settings.

Listing 1: General prompts used in PROF for all tasks.

```
1  You are an expert in robotics, reinforcement learning and code
       generation.
2  You are a reward engineer trying to write reward functions to solve
       reinforcement learning tasks as effective as possible.
3  Your goal is to write a reward function for the environment that
       will help the agent learn the task described in text.
4
5  Your reward function should use the current step's observation and
       action from the environment, as well as the next step's
       observation as input, and strictly follow the following format.
6  `
7  def compute_dense_reward(obs: np.ndarray, action: np.ndarray,
       next_obs: np.ndarray) -> float:
8      ...
9      return reward
10 `
11
12 The output of the reward function should be a weighted sum of
       multiple types of rewards.
13 The code output should be formatted as a python code string: "```
       python ... ```". Just the function body is fine.
14
15 Note:
16 1. Do not use information you are not given! Do not make assumptions
        about any unknown information! Do not print any logs!
17 2. Focus on the most relevant information.
18 3. The code should be as generic, complete and not contain omissions
        !
19 4. Avoid dividing by zero!
20 5. When you writing code, you can also add some comments as your
        thought.
21 6. You are allowed to use any existing python package if applicable.
        But only use these packages when it's really necessary.
22 7. The reward function code must be directly executable. It is not
        allowed to use undefined variables and methods, and it is not
        allowed to include unimplemented parts.
23
24 Tips:
25 1. If the robot needs to go to a target position, the reward can be
        constructed using the Euclidean distance between the current
        position and the target position.
26 2. The degree of goal completion is the most important factor in
        reward design. A higher degree of completion should correspond to
         a larger reward. In addition, it is possible to define several
        thresholds and provide bonus rewards when the degree of
        completion exceeds these thresholds.
27 3. The action penalty is a reasonable design choice, but the
        coefficient should not be too large.
28 4. To bound the velocity of bodies in the environment, a minor
        velocity penalty can be applied to the environment's full
        dynamics.
29 5. Positive rewards must be given for transitions that facilitate
        progress toward the goal, and penalties must be applied for
        transitions that hinder it. Do not reward only helpful
        transitions and ignore those that do not contribute.
30 6. Designing potential-based rewards is an effective method to
        structure learning signals. For example, instead of defining the
        reward based on the absolute distance to the target position, the
         reward can be constructed using the change in distance to the
        target position between the current step and the next step.
31
32 You should:
33 1. Give your thought about the task.
```

```
34  2. Think step by step and analyze positive and negative statuses or
        behaviors that can be reflected in which part of the observation
        and action.
35  3. Give a Python function that strictly follows the format mentioned
        previously.
```

## C.2   TASK-SPECIFIC PROMPTS

Task-specific prompts are constructed primarily from task and environment descriptions obtained from OpenAI official documentation[2]. The task description explains the task scenario, the acting agent, and the intended objective. Additional clarification is provided when the original descriptions are too brief to ensure clarity and completeness. When termination conditions are available in the source material, they are included in the prompt to discourage unsafe behaviors. In real-world scenarios, this information is also easily accessible because it is part of the data in the Markov Decision Process (MDP). The environment description provides a detailed description of the observation and action spaces while excluding irrelevant details such as variable names used in the corresponding XML file.

Task-specific prompts differ across environments due to variations in their state and action spaces. In the MuJoCo and Adroit domains, these prompts remain consistent across datasets of various quality within the same environment. In contrast, the AntMaze environment exhibits partial variability: while prompts of "large" (*large-diverse-v0*, *large-play-v0*) environments are consistent across dataset qualities, the "medium" (*medium-diverse-v0*, *medium-play-v0*) and "umaze" (*umaze-diverse-v0*, *umaze-v0*) environments employ slightly different prompts across datasets, as differences in their trajectory collection methods. The task-specific prompts for the *Half Cheetah*, *Hopper*, and *Walker2D* environments are presented in Listing 2, Listing 3, and Listing 4, respectively. For the comprehensive list of all task-specific prompts, please refer to the code in the supplementary material. Note that we modify these contents by removing certain spaces and hyphens from original prompts to ensure compatibility with the column width. These adjustments are minimal and are not expected to impact the experimental outcomes.

Listing 2: The task-specific prompts for *Half Cheetah* tasks.

```
1  ## Task Description
2  The environment description is:
3  The HalfCheetah is a 2-dimensional robot consisting of 9 body parts
       and 8 joints connecting them (including two paws). The goal is to
        apply torque to the joints to make the cheetah run forward (
       right) as fast as possible, with a positive reward based on the
       distance moved forward and a negative reward for moving backward.
        The cheetah's torso and head are fixed, and torque can only be
       applied to the other 6 joints over the front and back thighs (
       which connect to the torso), the shins (which connect to the
       thighs), and the feet (which connect to the shins).
4
5  ## Observation Space
6  The observation space of the environment is:
7
8  The observation space is a `Box(-Inf, Inf, (17,), float64)` where
       the elements are as follows:
9  | Num    | Observation                  | Min | Max | Type (Unit)
           |
10 |----------|----------------------------|-------|------|-------------------------|
11 | 0      | z-coordinate of the front tip | -Inf | Inf | position (m)
           |
12 | 1      | angle of the front tip       | -Inf | Inf | angle (rad)
           |
13 | 2      | angle of the back thigh      | -Inf | Inf | angle (rad)
           |
14 | 3      | angle of the back shin       | -Inf | Inf | angle (rad)
           |
```

[2]https://gymnasium.farama.org/

```
15 | | 4      | angle of the back foot        | -Inf | Inf | angle (rad)
          |
16 | | 5      | angle of the front thigh      | -Inf | Inf | angle (rad)
          |
17 | | 6      | angle of the front shin       | -Inf | Inf | angle (rad)
          |
18 | | 7      | angle of the front foot       | -Inf | Inf | angle (rad)
          |
19 | | 8      | velocity of the x-coordinate of front tip | -Inf | Inf |
       velocity (m/s) |
20 | | 9      | velocity of the z-coordinate of front tip | -Inf | Inf |
       velocity (m/s) |
21 | | 10     | angular velocity of the front tip | -Inf | Inf | angular
       velocity (rad/s) |
22 | | 11     | angular velocity of the back thigh | -Inf | Inf | angular
       velocity (rad/s) |
23 | | 12     | angular velocity of the back shin | -Inf | Inf | angular
       velocity (rad/s) |
24 | | 13     | angular velocity of the back foot | -Inf | Inf | angular
       velocity (rad/s) |
25 | | 14     | angular velocity of the front thigh | -Inf | Inf | angular
        velocity (rad/s) |
26 | | 15     | angular velocity of the front shin | -Inf | Inf | angular
       velocity (rad/s) |
27 | | 16     | angular velocity of the front foot | -Inf | Inf | angular
       velocity (rad/s) |
28 |
29 |
30 | ## Action Space
31 | The action space of the environment is:
32 |
33 | The action space is a `Box(-1, 1, (6,), float32)`. An action
       represents the torques applied at the hinge joints.
34 | | Num | Action                    | Control Min | Control Max |
       Type (Unit) |
35 | |-----|---------------------------|-------------|-------------|-----------------|
36 | | 0 | Torque applied on the back thigh rotor | -1 | 1    | torque (N
       m) |
37 | | 1 | Torque applied on the back shin rotor | -1 | 1    | torque (N
       m) |
38 | | 2 | Torque applied on the back foot rotor | -1 | 1    | torque (N
       m) |
39 | | 3 | Torque applied on the front thigh rotor | -1 | 1   | torque (N
       m) |
40 | | 4 | Torque applied on the front shin rotor | -1 | 1   | torque (N
       m) |
41 | | 5 | Torque applied on the front foot rotor | -1 | 1   | torque (N
       m) |
```

Listing 3: The task-specific prompts for *Hopper* tasks.

```
1 | ## Task Description
2 | The environment description is:
3 | The hopper is a two-dimensional one-legged figure consisting of four
       main body parts - the torso at the top, the thigh in the middle,
       the leg at the bottom, and a single foot on which the entire
     body rests. The goal is to make hops that move in the forward (
     right) direction by applying torque to the three hinges that
     connect the four body parts. The main component of the reward
     function is based on movement: moving forward yields positive
     rewards, while moving backward results in negative rewards. In
     addition, the hopper can be slightly encouraged to maintain a
     healthy posture and slightly penalized for unhealthy posture. The
      environment terminates when the hopper is unhealthy. The hopper
```

```
        is unhealthy if any of the following happens: (1) An element of `
        observation[1:]` is no longer contained in the closed interval
        [-100, 100]. (2) The height of the hopper (`observation[0]`) is
        no longer contained in the closed interval [0.7, +∞] (usually
        meaning that it has fallen). (3) The angle of the torso (`
        observation[1]`) is no longer contained in the closed interval
        [-0.2, 0.2].
4
5 ## Observation Space
6 The observation space of the environment is:
7
8 The observation space is a `Box(-Inf, Inf, (11,), float64)` where
        the elements are as follows:
9 | Num   | Observation                        | Min | Max | Type (
        Unit)       |
10 |---------|----------------------------|-------|-------|-------------------------|
11 | 0      | z-coordinate of the torso (height of hopper) | -Inf | Inf
        | position (m) |
12 | 1      | angle of the torso                     | -Inf | Inf | angle (
        rad)     |
13 | 2      | angle of the thigh joint            | -Inf | Inf | angle (
        rad)     |
14 | 3      | angle of the leg joint              | -Inf | Inf | angle (
        rad)     |
15 | 4      | angle of the foot joint             | -Inf | Inf | angle (
        rad)     |
16 | 5      | velocity of the x-coordinate of the torso | -Inf | Inf |
        velocity (m/s) |
17 | 6      | velocity of the z-coordinate (height) of torso | -Inf |
        Inf | velocity (m/s) |
18 | 7      | angular velocity of the angle of the torso | -Inf | Inf |
        angular velocity (rad/s) |
19 | 8      | angular velocity of the thigh hinge | -Inf | Inf | angular
         velocity (rad/s) |
20 | 9      | angular velocity of the leg hinge   | -Inf | Inf | angular
         velocity (rad/s) |
21 | 10     | angular velocity of the foot hinge  | -Inf | Inf | angular
         velocity (rad/s) |
22
23
24 ## Action Space
25 The action space of the environment is:
26
27 The action space is a `Box(-1, 1, (3,), float32)`. An action
        represents the torques applied at the hinge joints.
28 | Num | Action                      | Control Min | Control Max | Type (
        Unit) |
29 |-----|----------------------------|-------------|-------------|---------------|
30 | 0 | Torque applied on the thigh rotor | -1 | 1    | torque (N m) |
31 | 1 | Torque applied on the leg rotor | -1 | 1      | torque (N m) |
32 | 2 | Torque applied on the foot rotor | -1 | 1     | torque (N m) |
```

Listing 4: The task-specific prompts for *Walker2D* tasks.

```
1 ## Task Description
2 The environment description is:
3 The walker is a two-dimensional bipedal robot consisting of seven
        main body parts – a single torso at the top (with the two legs
        splitting after the torso), two thighs in the middle below the
        torso, two legs below the thighs, and two feet attached to the
        legs on which the entire body rests. The goal is to walk in the
        forward (right) direction by applying torque to the six hinges
        connecting the seven body parts. The main component of the reward
```

```
         function is based on movement: moving forward yields positive
        rewards, while moving backward results in negative rewards. In
        addition, the walker can be slightly encouraged to maintain a
        healthy posture and slightly penalized for unhealthy posture. The
         environment terminates when the walker is unhealthy. The walker
        is unhealthy if any of the following happens: (1) Any of the
        state space values is no longer finite. (2) The z-coordinate of
        the torso (the height) is not in the closed interval [0.8, 1.0].
         (3) The absolute value of the angle (`observation[1]`) is not in
        the closed interval [-1, 1].

## Observation Space
The observation space of the environment is:

The observation space is a `Box(-Inf, Inf, (17,), float64)` where
    the elements are as follows:
| Num    | Observation                          | Min | Max | Type (
    Unit)       |
|----------|---------------------------|-------|-------|--------------------------|

| 0      | z-coordinate of the torso (height of Walker2d) | -Inf |
    Inf | position (m) |
| 1      | angle of the torso                   | -Inf | Inf | angle (
    rad)      |
| 2      | angle of the thigh joint             | -Inf | Inf | angle (
    rad)      |
| 3      | angle of the leg joint               | -Inf | Inf | angle (
    rad)      |
| 4      | angle of the foot joint              | -Inf | Inf | angle (
    rad)      |
| 5      | angle of the left thigh joint        | -Inf | Inf | angle (
    rad)      |
| 6      | angle of the left leg joint          | -Inf | Inf | angle (
    rad)      |
| 7      | angle of the left foot joint         | -Inf | Inf | angle (
    rad)      |
| 8      | velocity of the x-coordinate of the torso | -Inf | Inf |
    velocity (m/s) |
| 9      | velocity of the z-coordinate (height) of torso | -Inf |
    Inf | velocity (m/s) |
| 10     | angular velocity of the angle of the torso | -Inf | Inf |
    angular velocity (rad/s) |
| 11     | angular velocity of the thigh hinge  | -Inf | Inf | angular
     velocity (rad/s) |
| 12     | angular velocity of the leg hinge    | -Inf | Inf | angular
    velocity (rad/s) |
| 13     | angular velocity of the foot hinge   | -Inf | Inf | angular
    velocity (rad/s) |
| 14     | angular velocity of the thigh hinge | -Inf | Inf | angular
    velocity (rad/s) |
| 15     | angular velocity of the leg hinge    | -Inf | Inf | angular
    velocity (rad/s) |
| 16     | angular velocity of the foot hinge   | -Inf | Inf | angular
    velocity (rad/s) |

## Action Space
The action space of the environment is:

The action space is a `Box(-1, 1, (6,), float32)`. An action
    represents the torques applied at the hinge joints.
| Num | Action                       | Control Min | Control Max | Type
     (Unit) |
|-----|------------------------------|-------------|-------------|-----------------|
```

```
36 | 0  | Torque applied on the thigh rotor | -1 | 1       | torque (N m
        ) |
37 | 1  | Torque applied on the leg rotor | -1  | 1        | torque (N m
        ) |
38 | 2  | Torque applied on the foot rotor | -1 | 1        | torque (N m
        ) |
39 | 3  | Torque applied on the left thigh rotor| -1 | 1   | torque (N m
        ) |
40 | 4  | Torque applied on the left leg rotor | -1 | 1     | torque (N m
        ) |
41 | 5  | Torque applied on the left foot rotor | -1 | 1    | torque (N m
        ) |
```

### C.3 LOSS FEEDBACK PROMPTS

We implement the loss prompt template $P_{\text{loss}}(x_g, x_t, R_{\arg\max_i S_i}, R_{\arg\min_i S_i})$ used in equation 10 following (Li et al., 2025), as detailed in Listing 5. The loss prompt template remains consistent across all experiments to ensure the generalization capability of PROF. The input {query} is constructed by concatenating the general prompts $x_g$ with the task-specific prompts $x_t$. The {chosen_response} corresponds to the reward function $R_{\arg\max_i S_i}$ with the highest dominance score, while the {rejected_response} corresponds to the reward function $R_{\arg\min_i S_i}$ with the lowest dominance score. These components jointly facilitate a loss feedback that enables LLMs to analyze preference relationships between reward functions. The definitions of $P_{\text{grad}}(\mathcal{L})$ and $P_{\text{update}}(\nabla_v \mathcal{L})$ used in equation 11 and equation 12 are consistent with those in TextGrad (Yuksekgonul et al., 2025). We utilize the official implementation of TextGrad, specifically at commit bf5b0c5[3].

Listing 5: Loss prompt template used in PROF for all tasks.

```
1 You are a language model tasked with evaluating a chosen response by
     comparing it with a rejected response to a user query. Analyze
     the strengths and weaknesses of each response, step by step, and
     explain why one is chosen or rejected.
2
3 **User Query**:
4 {query}
5
6 **Rejected Response**:
7 {rejected_response}
8
9 **Do NOT generate a response to the query. Be concise.** Below is
     the **Chosen Response**.
10 {chosen_response}
```

## D EXPERIMENTAL DETAILS

We conduct experiments on D4RL (Fu et al., 2020) datasets, including 9 MuJoCo "v2" datasets (*halfcheetah-medium*, *halfcheetah-medium-replay*, *halfcheetah-medium-expert*, *hopper-medium*, *hopper-medium-replay*, *hopper-medium-expert*, *walker2d-medium*, *walker2d-medium-replay*, *walker2d-medium-expert*), 6 AntMaze "v0" datasets (*umaze*, *umaze-diverse*, *medium-diverse*, *medium-play*, *large-diverse*, and *large-play*), and 8 Adroit "v0" datasets (*pen-human*, *pen-cloned*, *door-human*, *door-cloned*, *relocate-human*, *relocate-cloned*, *hammer-human*, and *hammer-cloned*). We adopt TD3_BC (Fujimoto & Gu, 2021) and IQL (Kostrikov et al., 2022) as the base offline RL algorithms. To ensure fair comparisons, the hyperparameters strictly follow those used in the SEABO paper. We report the hyperparameter settings for IQL and TD3_BC in Table 5. The shared parameters for LLM API queries across all experiments are provided in Table 6. PROF designs reward functions for all tasks using $(s, a, s')$, and we conduct experiments using $(s, s')$ in Appendix E.4.

---

[3] https://github.com/zou-group/textgrad

Table 5: Hyperparameters of IQL and TD3_BC across across domains. Domain-specific values are shown in parentheses.

|  | Hyperparameter | Value (Domain) |
|---|---|---|
| Shared Configurations | Hidden layers | $(256, 256)$ |
|  | Discount factor | $0.99$ |
|  | Actor learning rate | $3 \times 10^{-4}$ |
|  | Critic learning rate | $3 \times 10^{-4}$ |
|  | Batch size | $256$ |
|  | Optimizer | Adam (Kingma & Ba, 2014) |
|  | Target update rate | $5 \times 10^{-3}$ |
|  | Activation function | ReLU |
| IQL | Value learning rate | $3 \times 10^{-4}$ (MuJoCo) |
|  | Temperature | 3.0 (MuJoCo), 10.0 (AntMaze), 0.5 (Adroit) |
|  | Expectile | 0.7 (MuJoCo, Adroit), 0.9 (AntMaze) |
|  | Actor dropout rate | NA (MuJoCo, Adroit), 0.1 (Adroit) |
| TD3_BC | Policy noise | 0.2 (MuJoCo) |
|  | Policy noise clipping | $(-0.5, 0.5)$ (MuJoCo) |
|  | Policy update frequency | 2 (MuJoCo) |
|  | Normalization weight | 2.5 (MuJoCo) |

Table 6: Hyperparameter configuration for LLM API queries.

| Hyperparameter | Value |
|---|---|
| Temperature | 0.7 |
| Max output tokens | 10000 |
| Top-p | 1.0 |

We follow the previous works Luo et al. (2023); Lyu et al. (2024) to obtain expert demonstrations in order to ensure a fair comparison between different algorithms. Specifically, we select the trajectory with the highest return as expert demonstrations on MuJoCo locomotion tasks and Adroit tasks, and we filter the trajectory that reaches the goal in AntMaze tasks.

The results of baselines are directly obtained from the SEABO paper. Our implementations of IQL and TD3_BC leverage the official SEABO codebase[4]. We adopt the normalized score metric as proposed in the D4RL (Fu et al., 2020), which has been widely employed in prior works (Luo et al., 2023; Lyu et al., 2024). Let $J$ denote the average return achieved by the learned policy in the test environments. The normalized score is defined as:

$$\text{Normalized Score} = \frac{J - J_R}{J_E - J_R} \times 100$$

where $J_R$ and $J_E$ represent the average returns of a random and an expert policy, respectively. Under this formulation, a score of 0 corresponds to the performance of a random policy, while a score of 100 indicates expert-level performance.

To constrain the range of rewards, prior approaches have commonly applied squashing functions. However, these methods often lack standardization, employing different squashing functions for different environments (Luo et al., 2023) or varying the scaling factors when using the same function (Lyu et al., 2024). On the other hand, we do not use the $1000/(\max\_\text{return} - \min\_\text{return})$ reward scaling method like IQL, as the reward functions generated by LLMs are diverse and the difference in $\max\_\text{return} - \min\_\text{return}$ may be large. In contrast, we adopt a unified and domain-agnostic strategy based on simple min-max normalization, which provides effective and consistent reward constraints across tasks. Specifically, we linearly rescale the reward values into the range $[R_{\min}, R_{\max}]$ as follows:

$$\hat{r} = R_{\min} + \frac{(r - r_{\min})(R_{\max} - R_{\min})}{r_{\max} - r_{\min}},$$

---

[4] https://github.com/dmksjfl/SEABO

where $r$ is the original reward, $r_{\min}$ and $r_{\max}$ denote the minimum and maximum reward values in the dataset, $R_{\min}$ and $R_{\max}$ are scaling bound hyperparameters, and $\hat{r}$ is the normalized reward. The default scaling bound hyperparameters used in PROF are detailed in Table 7. For a few tasks, we slightly adjust the hyperparameters to achieve better performance. In the case of "IQL+PROF" with reward design based on $(s, a, s')$, we use $R_{\max} = 1$ for *hopper-medium-expert*, $(R_{\min}, R_{\max}) = (0, 4)$ for *pen-human*, $(R_{\min}, R_{\max}) = (-5, 0)$ for *antmaze-umaze-diverse-v0* and *antmaze-umaze-v0*. For "TD3_BC+PROF" with reward design using $(s, a, s')$, we set $(R_{\min}, R_{\max}) = (0, 0.5)$ for *halfcheetah-medium-expert*. When using "IQL+PROF" with reward design based on $(s, s')$, we apply $R_{\max} = 1$ for both *halfcheetah-medium-expert* and *hopper-medium-expert*.

Table 7: Default reward scaling settings for IQL and TD3_BC across various tasks.

| Algorithm | Task Domain | $R_{\min}$ | $R_{\max}$ |
|---|---|---|---|
| IQL | MuJoCo | 0 | 2 |
| | AntMaze and Adroit | -2 | 0 |
| TD3_BC | MuJoCo | -1 | 1 |

Our method for selecting expert trajectories aligns with prior works (Luo et al., 2023; Lyu et al., 2024). For MuJoCo and Adroit tasks, we define the trajectory with the highest return as the expert demonstration. For AntMaze tasks, we define the trajectory that successfully accomplishes the goal as the expert demonstration.

All experiments are conducted using `mujoco-py` version `1.50.1.68`, Gym version `0.18.3`, and `PyTorch` version `1.8`. Tasks in the Adroit domain are executed on a single NVIDIA RTX 3090 GPU paired with an AMD EPYC 7452 32-core processor. All other tasks utilized a single NVIDIA RTX 4090 GPU with an AMD EPYC 9554 64-core processor. For each task, PROF constructs $H = 10^4$ noisy trajectories, which are reused throughout the Reward Preference Ranking process. Both the construction of noisy trajectories and the computation of the dominance score for each candidate reward function require only a few minutes. Therefore, excluding the latency introduced by LLM queries, which depends primarily on network conditions and usage limits, the overall computational cost of PROF remains low and within an acceptable range.

# E ADDITIONAL RESULTS

## E.1 RESULTS ON ADROIT DOMAIN

We evaluate the performance of PROF on the challenging Adroit domain. Experiments are conducted on 4 tasks: *pen*, *door*, *relocate*, and *hammer*, each paired with two "v0" dataset types, *human* and *cloned*, resulting in a total of 8 evaluation settings. Table 8 reports the mean D4RL normalized scores achieved by various algorithms, along with their improvement ratios relative to IQL trained with ground-truth rewards. Results show that PROF achieves the highest improvement ratio in 6 out of 8 tasks, demonstrating strong performance across complex control problems. Moreover, PROF achieves the highest average improvement across all tasks, with a 102.3% gain over IQL, clearly outperforming SEABO, which achieves 52.2%. In contrast, OTR fails to recover the performance of IQL, exhibiting a 5.7% degradation. While PROF slightly underperforms SEABO on the *pen human* and *pen cloned* tasks, leading to a marginally lower total score, we attribute this to the suitability of imitating expert demonstrations for these particular tasks.

## E.2 COMPLETE ITERATIVE PERFORMANCE CHANGES

Table 9 presents the performance of PROF across different iteration numbers on three domains: MuJoCo, AntMaze, and Adroit. In all domains, the total score initially increases before declining. The results indicate that a single iterative optimization is sufficient for simpler tasks, while more complex tasks benefit from two iterations. Further optimization beyond this point appears to degrade performance, suggesting that over-optimization of the reward functions should be avoided.

Table 10 reports the total token consumption per iteration for $n = 5$ parallel samplings. At $T = 0$, corresponding to the initial reward function generation using general prompts and task-specific

Table 8: **Comparison of PROF and baselines on D4RL Adroit tasks.** We report the mean D4RL normalized score with standard deviation, calculated across 5 random seeds. For each task, the value in parentheses denotes the relative improvement of the "IQL+" algorithms compared to the IQL trained with ground-truth rewards. For the "total" row, the value in parentheses indicates the average relative improvement across all tasks. We bold and shade the cells with the highest improvement percentage in each task.

| Task | IQL | IQL+OTR | IQL+SEABO | IQL+PROF |
|------|-----|---------|-----------|----------|
| pen-human | 70.7±8.6 | 66.8±21.2 (-5.5%) | **94.3±12.0 (+33.4%)** | 85.8±17.4 (+21.4%) |
| pen-cloned | 37.2±7.3 | 46.9±20.9 (+26.1%) | **48.7±15.3 (+30.9%)** | 46.7±16.6 (+25.5%) |
| door-human | 3.3±1.3 | 5.9±2.7 (+78.8%) | 5.1±2.0 (+54.5%) | **8.1±3.9 (+145.5%)** |
| door-cloned | 1.6±0.5 | 0.0±0.0 (-100.0%) | 0.4±0.8 (-75.0%) | **1.1±2.0 (-31.3%)** |
| relocate-human | 0.1±0.0 | 0.1±0.1 (+0.0%) | 0.4±0.5 (+300.0%) | **0.5±0.6 (+400.0%)** |
| relocate-cloned | -0.2±0.0 | **-0.2±0.0 (+0.0%)** | **-0.2±0.0 (+0.0%)** | **-0.2±0.0 (+0.0%)** |
| hammer-human | 1.6±0.6 | 1.8±1.4 (+12.5%) | 2.7±1.8 (+68.8%) | **4.8±3.1 (+200.0%)** |
| hammer-cloned | 2.1±1.0 | 0.9±0.3 (-57.1%) | 2.2±0.8 (+4.8%) | **3.3±2.5 (+57.1%)** |
| total | 116.4 | 122.2 (-5.7%) | 153.6 (+52.2%) | **150.1 (+102.3%)** |

prompts, the token usage remains relatively low despite the parallel samplings. For $T \in \{1, 2, 3\}$, TextGrad performs loss computation, backpropagation, and code updates in sequence. As a result, the total token consumption across the 5 independent executions is significantly higher. The slight increase in token consumption per iteration as $T$ grows is due to the progressively more complex reward functions generated by the LLM. Note that the column labeled **Total Usage** reports the token consumption for a full run of PROF with $T = 3$. Using the reasonable number of iterations, specifically $T = 1$ for simple tasks and $T = 2$ for complex tasks, substantially reduces the actual token consumption.

### E.3  COMPARISON OF PROF AND IMITATION LEARNING ALGORITHMS (Q4)

To further validate the effectiveness of PROF, we compare it against recent strong offline IL methods under the same settings as SEABO. The baselines include: (i) **SQIL** (Reddy et al., 2020), which assigns a reward of $+1$ to expert transitions and $0$ otherwise. (ii) **DemoDICE** (Kim et al., 2022b), an algorithm designed to utilize imperfect demonstrations for offline IL. (iii) **SMODICE** (Ma et al., 2022), a regression-based offline IL algorithm derived through the principle of state-occupancy matching. (iv) **PWIL** (Dadashi et al., 2021), imitation learning using the Wasserstein distance between expert and agent state-action distributions. Although SQIL and PWIL are originally proposed as online IL algorithms, SEABO adapts them to the offline setting by replacing the base algorithm in SQIL with TD3+BC and using IQL as the base algorithm for PWIL. In addition, SEABO modifies SMODICE by incorporating action information during discriminator training. We report the baseline results directly from SEABO paper. All settings remain consistent with Section 5.1, and both SEABO and PROF employ IQL as the base RL algorithm. The results are presented in Table 11. We observe that: **Obs.❼ PROF consistently outperforms or matches strong imitation learning baselines across all tasks.** Notably, it achieves a substantial improvement in the overall score, indicating its effectiveness in modeling the reward function distribution. These findings highlight the superiority of PROF over imitation learning approaches.

### E.4  EXPERIMENTS ON THE STATE-ONLY SETTING (Q5)

We further evaluate PROF in a state-only setting, where only state transitions $(s, s')$ are available, without access to action information $a$. We compare our method against **SMODICE** (Ma et al., 2022), **OTR** (Luo et al., 2023), and **SEABO** (Lyu et al., 2024). We also consider two additional baselines: (i) **LobsDICE**, which learns to imitate expert policies by optimizing in the stationary distribution space. (ii) **PWIL-state** (Lyu et al., 2024), a modified version of PWIL (Dadashi et al., 2021) that relies solely on observations to compute rewards. Results for all baselines are sourced directly from the SEABO paper. For PROF, experimental configurations remain consistent with Section 5.1, except that the prompt provided to the LLMs is modified to ensure that the reward function is conditioned on $(s, s')$ rather than $(s, a, s')$. Table 12 summarizes the comparative results.

Table 9: **Detailed comparison of baselines and PROF using different $T$ on D4RL MuJoCo, AntMaze and Adroit tasks.** We report the mean D4RL normalized score with standard deviation, calculated across 5 random seeds. We shade the cells with the highest total scores in PROF with different iteration numbers $T \in \{0, 1, 2, 3\}$. Abbreviations: hc = halfcheetah, hop = hopper, w2d = walker2d; m = medium, mr = medium-replay, me = medium-expert.

| Task | IQL | IQL+SEABO | PROF (T=0) | PROF (T=1) | PROF (T=2) | PROF (T=3) |
|---|---|---|---|---|---|---|
| hc-m | 47.4±0.2 | 44.8±0.3 | 47.6±0.2 | 47.4±0.1 | 45.2±0.2 | 44.9±0.2 |
| hop-m | 66.2±5.7 | 80.9±3.2 | 64.3±4.0 | 65.9±3.8 | 71.2±4.9 | 59.4±3.8 |
| w2d-m | 78.3±8.7 | 80.9±0.6 | 83.3±0.6 | 82.2±1.2 | 82.2±1.4 | 82.3±2.5 |
| hc-mr | 44.2±1.2 | 42.3±0.1 | 43.9±1.1 | 42.6±2.2 | 44.2±0.6 | 43.1±2.4 |
| hop-mr | 94.7±8.6 | 92.7±2.9 | 91.2±6.1 | 96.6±3.1 | 93.9±3.9 | 93.9±3.9 |
| w2d-mr | 73.8±7.1 | 74.0±2.7 | 64.8±15.6 | 82.4±1.8 | 78.1±4.7 | 77.8±2.8 |
| hc-me | 86.7±5.3 | 89.3±2.5 | 89.5±3.0 | 89.6±3.2 | 90.5±4.5 | 90.7±2.9 |
| hop-me | 91.5±14.3 | 97.5±5.8 | 92.4±11.3 | 108.3±5.1 | 104.6±13.1 | 98.5±14.3 |
| w2d-me | 109.6±1.0 | 110.9±0.2 | 109.4±0.7 | 109.8±0.7 | 109.2±0.4 | 109.4±0.5 |
| total (MuJoCo) | 692.4 | 713.3 | 686.4 | **724.8** | 719.1 | 700.0 |
| umaze | 87.5±2.6 | 90.0±1.8 | 93.0±3.9 | 93.0±3.9 | 93.0±3.9 | 93.0±3.9 |
| umaze-diverse | 62.2±13.8 | 66.2±7.2 | 59.2±12.5 | 69.0±9.1 | 69.0±9.1 | 69.0±9.1 |
| medium-diverse | 70.0±10.9 | 72.2±4.1 | 73.8±4.2 | 73.8±4.2 | 75.8±5.8 | 71.0±3.8 |
| medium-play | 71.2±7.3 | 71.6±5.4 | 76.8±3.7 | 75.0±6.2 | 76.6±3.3 | 76.6±3.3 |
| large-diverse | 47.5±9.5 | 50.0±6.8 | 51.6±4.5 | 51.6±4.5 | 51.6±4.5 | 51.6±4.5 |
| large-play | 39.6±5.8 | 50.8±8.7 | 43.4±10.9 | 43.4±10.9 | 43.4±10.9 | 43.2±3.1 |
| total (AntMaze) | 378.0 | 400.8 | 397.8 | 405.8 | **409.4** | 404.4 |
| pen-human | 70.7±8.6 | 94.3±12.0 | 77.8±13.5 | 71.3±18.6 | 85.8±17.4 | 76.5±19.3 |
| pen-cloned | 37.2±7.3 | 48.7±15.3 | 42.7±4.6 | 39.7±16.5 | 46.7±16.6 | 45.6±15.1 |
| door-human | 3.3±1.3 | 5.1±2.0 | 3.1±2.3 | 7.1±3.1 | 8.1±3.9 | 2.9±1.1 |
| door-cloned | 1.6±0.5 | 0.4±0.8 | 0.3±0.6 | 1.0±1.7 | 1.1±2.0 | 1.1±1.8 |
| relocate-human | 0.1±0.0 | 0.4±0.5 | 0.1±0.0 | 0.1±0.0 | 0.5±0.6 | 0.2±0.1 |
| relocate-cloned | -0.2±0.0 | -0.2±0.0 | -0.2±0.0 | -0.2±0.0 | -0.2±0.0 | -0.2±0.0 |
| hammer-human | 1.6±0.6 | 2.7±1.8 | 2.1±1.3 | 2.1±1.1 | 4.8±3.1 | 2.1±0.9 |
| hammer-cloned | 2.1±1.0 | 2.2±0.8 | 3.3±2.5 | 3.3±2.5 | 3.3±2.5 | 2.4±0.6 |
| total (Adroit) | 116.4 | 153.6 | 129.2 | 124.4 | **150.1** | 130.6 |
| total (All) | 1186.8 | 1267.7 | 1213.4 | 1255.0 | **1278.6** | 1235.0 |

The results show that: **Obs.⑥ PROF achieves the highest total score in the state-only setting**. Notably, no single method consistently outperforms others across all environments, only PROF and SEABO attain top performance on 4 out of 9 tasks, respectively. On the remaining tasks, PROF lags behind the best-performing approach on *hopper-medium* and *walker2d-medium-replay*, while exhibiting comparable performance on the others. These results demonstrate the effectiveness and generalization capabilities of PROF. However, the existence of a universally dominant algorithm in the state-only setting remains an open research question.

## F  COMPARISON WITH THE GROUND-TRUTH REWARD FUNCTIONS

To understand why PROF surpasses ground-truth rewards, we analyze representative environments from the MuJoCo, AntMaze, and Adroit domains. Specifically, we focus on the "v2" dateset `walker2d-medium-replay`, the "v0" datasets `antmaze-medium-diverse` and `door-human`. On these tasks, our approach consistently outperforms IQL trained with ground-truth rewards. As defined in the D4RL (Fu et al., 2020), the ground-truth reward functions for the `Walker2D` and `Door` tasks are detailed in Listing 6 and Listing 7, respectively. For `AntMaze`, the ground-truth is a sparse signal: a reward of $+1$ is given if task success, with zero reward otherwise. Reward functions designed by PROF are shown in Listing 8, Listing 9, and Listing 10. For `walker2d-medium-replay`, we present results using iteration $T = 1$, while for `antmaze-medium-diverse` and `door-human`, we report results using iteration $T = 2$.

Table 10: **Token usage of PROF using different $T$ on D4RL MuJoCo, AntMaze and Adroit tasks.** We report the total tokens consumed by sampling $n = 5$ in parallel at iterations $T \in \{0, 1, 2, 3\}$, respectively. The column labeled **Total Usage** denotes the cumulative tokens consumed when executing PROF fully with $T = 3$ for each task. Abbreviations: hc = halfcheetah, hop = hopper, w2d = walker2d; m = medium, mr = medium-replay, me = medium-expert.

| Token | PROF (T=0) | PROF (T=1) | PROF (T=2) | PROF (T=3) | Total Usage |
|---|---|---|---|---|---|
| hc-m | 11405 | 72486 | 76969 | 79365 | 240225 |
| hop-m | 11104 | 74940 | 90109 | 89820 | 265973 |
| w2d-m | 12756 | 75029 | 93202 | 97440 | 275546 |
| hc-mr | 11626 | 74798 | 80486 | 81627 | 250708 |
| hop-mr | 11364 | 77249 | 79791 | 89474 | 257878 |
| w2d-mr | 13048 | 77463 | 86976 | 88897 | 266386 |
| hc-me | 11886 | 73952 | 77418 | 79512 | 242816 |
| hop-me | 11215 | 77201 | 85790 | 90912 | 265118 |
| w2d-me | 13133 | 78769 | 80735 | 88200 | 260837 |
| Average (MuJoCo) | 11948 | 75765 | 83177 | 87496 | 258387 |
| umaze | 15695 | 91440 | 90969 | 91148 | 289252 |
| umaze-diverse | 15765 | 89989 | 99331 | 97143 | 302228 |
| medium-diverse | 16233 | 95103 | 101647 | 108562 | 321545 |
| medium-play | 15732 | 93650 | 99675 | 102524 | 311581 |
| large-diverse | 15444 | 90258 | 89738 | 87819 | 283259 |
| large-play | 15775 | 88870 | 93725 | 90230 | 288600 |
| Average (AntMaze) | 15774 | 91552 | 95848 | 96238 | 299411 |
| pen-human | 21494 | 106836 | 119344 | 120830 | 368504 |
| pen-cloned | 21746 | 113910 | 113693 | 121695 | 365913 |
| door-human | 20818 | 102553 | 105252 | 104768 | 333391 |
| door-cloned | 20958 | 108673 | 108863 | 114486 | 352990 |
| relocate-human | 21213 | 106046 | 115724 | 123810 | 366800 |
| relocate-cloned | 20839 | 99970 | 112901 | 116506 | 350216 |
| hammer-human | 21950 | 106768 | 117592 | 122441 | 368751 |
| hammer-cloned | 21835 | 114002 | 112861 | 113968 | 362666 |
| Average (Adroit) | 21357 | 106744 | 113238 | 117315 | 358654 |

Table 11: **Comparison of PROF and imitation learning algorithms on D4RL MuJoCo locomotion tasks.** We report the mean D4RL normalized score with standard deviation, calculated across 5 random seeds. We bold and shade the cells with the highest scores in each task. Abbreviations: hc = halfcheetah, hop = hopper, w2d = walker2d; m = medium, mr = medium-replay, me = medium-expert.

| Task Name | SQIL | DemoDICE | SMODICE | PWIL | PROF |
|---|---|---|---|---|---|
| hc-m | 31.3±1.8 | 42.5±1.7 | 41.7±1.0 | 44.4±0.2 | **47.4±0.1** |
| hop-m | 44.7±20.1 | 55.1±3.3 | 56.3±2.3 | 60.4±1.8 | **65.9±3.8** |
| w2d-m | 59.6±7.5 | 73.4±2.6 | 13.3±9.2 | 72.6±6.3 | **82.2±1.2** |
| hc-mr | 29.3±2.2 | 38.1±2.7 | 38.7±2.4 | **42.6±0.5** | **42.6±2.2** |
| hop-mr | 45.2±23.1 | 39.0±15.4 | 44.3±19.7 | 94.0±7.0 | **96.6±3.1** |
| w2d-mr | 36.3±13.2 | 52.2±13.1 | 44.6±23.4 | 41.9±6.0 | **82.4±1.8** |
| hc-me | 40.1±6.4 | 85.8±5.7 | 87.9±5.8 | 89.5±3.6 | **89.6±3.2** |
| hop-me | 49.8±5.8 | 92.3±14.2 | 76.0±8.6 | 70.9±35.1 | **108.3±5.1** |
| w2d-me | 35.9±22.2 | 106.9±1.9 | 47.8±31.1 | **109.8±0.2** | **109.8±0.7** |
| total | 372.2 | 585.3 | 450.6 | 626.1 | **724.8** |

The results show that the ground-truth reward for *walker2d-medium-replay* consists of three components: encouraging forward movement, encouraging survival, and applying an action regularization penalty. In contrast, PROF constructs a more complex reward function. In addition to encouraging forward movement and applying an action regularization penalty, it penalizes both excessively high

Table 12: **Comparison of PROF and baselines on D4RL MuJoCo locomotion tasks.** All algorithms are evaluated in a state-only setting, using only observations $(s, s')$ without access to actions $a$. PWIL-state indicates that PWIL uses only observations to compute rewards. We report the mean D4RL normalized score with standard deviation, calculated across 5 random seeds. We bold and shade the cells with the highest scores in each task. Abbreviations: hc = halfcheetah, hop = hopper, w2d = walker2d; m = medium, mr = medium-replay, me = medium-expert.

| Task | SMODICE | LobsDICE | PWIL-state | OTR | SEABO | PROF |
|------|---------|----------|------------|-----|-------|------|
| hc-m | 41.1±2.1 | 41.5±1.8 | 0.1±0.6 | 43.3±0.2 | **45.0±0.2** | 44.8±0.2 |
| hop-m | 56.5±1.8 | 56.9±1.4 | 1.4±0.5 | **78.7±5.5** | 74.7±5.2 | 71.2±2.5 |
| w2d-m | 15.5±18.6 | 69.3±5.4 | 0.2±0.2 | 79.4±1.4 | **81.3±1.3** | 81.2±2.4 |
| hc-mr | 39.2±3.1 | 39.9±3.1 | -2.4±0.2 | 41.3±0.6 | 42.4±0.6 | **45.1±0.5** |
| hop-mr | 55.3±21.4 | 41.6±16.8 | 0.7±0.2 | 84.8±2.6 | 88.0±0.7 | **93.2±9.5** |
| w2d-mr | 37.8±10.2 | 33.2±7.0 | -0.2±0.2 | 66.0±6.7 | **76.4±3.0** | 68.0±8.1 |
| hc-me | 88.0±4.0 | 89.4±3.2 | 0.0±1.0 | 89.6±3.0 | 91.8±1.5 | **92.9±0.5** |
| hop-me | 75.1±11.7 | 53.4±3.2 | 2.7±2.1 | 93.2±20.6 | 97.5±6.4 | **110.1±2.5** |
| w2d-me | 32.3±14.7 | 106.6±2.7 | 0.2±0.3 | 109.3±0.8 | **110.5±0.3** | 110.2±0.9 |
| total | 440.8 | 531.8 | 2.7 | 685.6 | 707.6 | **716.7** |

or low torso heights and extreme torso angles, encourages smooth acceleration, and penalizes rapid oscillations and abrupt changes in action. For *antmaze-medium-diverse*, the ground-truth reward is sparse. PROF designs a dense reward function that encourages forward movement and reaching the target while penalizing unhealthy postures, action regularization, and excessive movement speed. The ground-truth reward for *door-human* includes a penalty for the distance to the handle, a penalty for the door not being opened sufficiently, a speed penalty, and a segmented reward based on the angular position of the door hinge. PROF extends this reward structure by adding components for latch opening and action regularization.

The code-level analysis clearly shows the advantages of our approach. Specifically, PROF employs parallel sampling and iterative optimization to generate reward candidates that are both more complex and comprehensive. Moreover, the LLMs learn to exploit the difference between $s$ and $s'$ for effective reward shaping. PROF further incorporates Reward Preference Ranking (RPR), enabling the selection of reward functions whose distribution is most closely aligned with expert intention.

Listing 6: Ground-truth reward function of environment *Walker2D* defined in D4RL.

```python
def step(self, a):
    posbefore = self.sim.data.qpos[0]
    self.do_simulation(a, self.frame_skip)
    posafter, height, ang = self.sim.data.qpos[0:3]
    alive_bonus = 1.0
    reward = ((posafter - posbefore) / self.dt)
    reward += alive_bonus
    reward -= 1e-3 * np.square(a).sum()
    done = not (height > 0.8 and height < 2.0 and
                ang > -1.0 and ang < 1.0)
    ob = self._get_obs()
    return ob, reward, done, {}
```

Listing 7: Ground-truth reward function of environment *Door* defined in D4RL.

```python
def step(self, a):
    a = np.clip(a, -1.0, 1.0)
    try:
        a = self.act_mid + a*self.act_rng # mean center and scale
    except:
        a = a                       # only for the initialization phase
    self.do_simulation(a, self.frame_skip)
    ob = self.get_obs()
    handle_pos = self.data.site_xpos[self.handle_sid].ravel()
    palm_pos = self.data.site_xpos[self.grasp_sid].ravel()
    door_pos = self.data.qpos[self.door_hinge_did]
```

```
13      # get to handle
14      reward = -0.1*np.linalg.norm(palm_pos-handle_pos)
15      # open door
16      reward += -0.1*(door_pos - 1.57)*(door_pos - 1.57)
17      # velocity cost
18      reward += -1e-5*np.sum(self.data.qvel**2)
19
20      if ADD_BONUS_REWARDS:
21          # Bonus
22          if door_pos > 0.2:
23              reward += 2
24          if door_pos > 1.0:
25              reward += 8
26          if door_pos > 1.35:
27              reward += 10
28
29      goal_achieved = True if door_pos >= 1.35 else False
30
31      return ob, reward, False, dict(goal_achieved=goal_achieved)
```

Listing 8: Reward function of "v2" dataset *walker2d-medium-replay* designed by PROF using $T = 1$.

```
1  import numpy as np
2
3  def compute_dense_reward(obs: np.ndarray, action: np.ndarray,
     next_obs: np.ndarray) -> float:
4      # Initialize reward
5      reward = 0.0
6
7      # 1. Forward movement reward (primary goal)
8      forward_velocity = next_obs[8]
9      reward += 5.0 * forward_velocity # Adjusted weight for forward
         movement reward to balance other components
10
11     # 2. Posture maintenance
12     # 2.1. Penalize deviation from desired torso height [0.8, 1.0]
         with smoother penalties
13     torso_height = next_obs[0]
14     if torso_height < 0.8:
15         reward -= 5.0 * (0.8 - torso_height) ** 2 # Penalize for being
             too low
16     elif torso_height > 1.0:
17         reward -= 5.0 * (torso_height - 1.0) ** 2 # Penalize for being
             too high
18
19     # 2.2. Penalize deviation from desired torso angle [-1, 1] with a
         piecewise quadratic penalty
20     torso_angle = next_obs[1]
21     if torso_angle < -1.0:
22         reward -= 3.0 * (-1.0 - torso_angle) ** 2 # Penalize for
             extreme backward lean
23     elif torso_angle > 1.0:
24         reward -= 3.0 * (torso_angle - 1.0) ** 2 # Penalize for
             extreme forward lean
25
26     # 3. Action penalty (encourage smooth and efficient movements)
27     reward -= 0.05 * np.sum(np.square(action)) # Increased weight for
         action penalty to discourage excessive torque
28
29     # 4. Encourage smooth progress (potential-based reward with
         efficiency considerations)
30     delta_x = next_obs[8] - obs[8] # Change in x-coordinate (progress
         )
31     reward += 2.0 * delta_x # Reward consistent progress
```

```
32    velocity_smoothness_penalty = np.abs(forward_velocity - (obs[8] /
          2)) # Penalize large oscillations in velocity
33    reward -= 1.0 * velocity_smoothness_penalty
34
35    # 5. Penalize unsafe behaviors (e.g., rapid oscillations or
          abrupt changes)
36    joint_velocity_penalty = 0.01 * np.sum(np.abs(next_obs[10:])) #
          Penalize rapid joint oscillations
37    action_smoothness_penalty = 0.01 * np.sum(np.abs(action - np.mean
          (action))) # Penalize abrupt changes in actions
38    reward -= joint_velocity_penalty + action_smoothness_penalty
39
40    return reward
```

Listing 9: Reward function of "v0" dataset *antmaze-medium-diverse* designed by PROF using $T = 2$.

```
1  import numpy as np
2
3  def compute_dense_reward(
4      obs: np.ndarray,
5      action: np.ndarray,
6      next_obs: np.ndarray,
7      forward_weight: float = 5.0,
8      goal_weight: float = 3.0,
9      posture_penalty_weight: float = -10.0,
10     action_penalty_weight: float = -0.01,
11     velocity_penalty_weight: float = -0.005,
12     z_target: float = 0.6,
13     z_tolerance: float = 0.01,
14     velocity_clip: float = 10.0
15 ) -> float:
16     """
17     Computes the dense reward for the RL environment, considering
           progress toward the goal,
18     efficient movements, healthy posture, and stability.
19
20     Args:
21         obs (np.ndarray): Current observation.
22         action (np.ndarray): Action taken.
23         next_obs (np.ndarray): Next observation.
24         forward_weight (float): Weight for the forward movement reward
               .
25         goal_weight (float): Weight for the goal-reaching reward.
26         posture_penalty_weight (float): Weight for the posture penalty
               .
27         action_penalty_weight (float): Weight for the action penalty.
28         velocity_penalty_weight (float): Weight for the velocity
               penalty.
29         z_target (float): Target height for the torso.
30         z_tolerance (float): Tolerance for the posture penalty.
31         velocity_clip (float): Maximum velocity value for clipping.
32
33     Returns:
34         float: The computed reward.
35     """
36     # Extract relevant variables from the observations
37     x_pos, y_pos, z_pos = obs[0], obs[1], obs[2] # Current position
38     next_x_pos, next_y_pos, next_z_pos = next_obs[0], next_obs[1],
           next_obs[2] # Next position
39     x_vel, y_vel = np.clip(obs[15], -velocity_clip, velocity_clip),
           np.clip(obs[16], -velocity_clip, velocity_clip) # Clipped
           velocities
40     goal_x, goal_y = obs[29], obs[30] # Goal position
41
```

```python
42    # Extract actions for penalty
43    torque_penalty = np.sum(np.square(action)) # Sum of squared
          torques (penalize large actions)
44    torque_std_penalty = np.std(action) # Penalize uneven torque
          application
45
46    # Compute distances to the goal
47    current_goal_dist = np.sqrt((goal_x - x_pos)**2 + (goal_y - y_pos
          )**2) # Current distance to goal
48    next_goal_dist = np.sqrt((goal_x - next_x_pos)**2 + (goal_y -
          next_y_pos)**2) # Next distance to goal
49
50    # Reward components
51    # 1. Directional reward for moving forward
52    forward_direction = np.array([1.0, 0.0]) # Desired forward
          direction along the x-axis
53    movement_vector = np.array([next_x_pos - x_pos, next_y_pos -
          y_pos])
54    forward_reward = np.dot(movement_vector, forward_direction) #
          Reward for moving in the desired direction
55
56    # 2. Goal-reaching reward (potential-based reward: reduction in
          distance to goal)
57    initial_goal_dist = np.sqrt((goal_x - obs[0])**2 + (goal_y - obs
          [1])**2) # Initial distance to goal
58    goal_reward = ((current_goal_dist - next_goal_dist) /
          initial_goal_dist) if initial_goal_dist > 0 else 0.0
59
60    # 3. Posture penalty (encourage healthy z-pos in range [0.2,
          1.0])
61    if next_z_pos < (0.2 - z_tolerance) or next_z_pos > (1.0 +
          z_tolerance):
62      posture_penalty = posture_penalty_weight # Strong penalty for
            unhealthy posture
63    else:
64      posture_penalty = -abs(next_z_pos - z_target) # Reward for
            staying near the target height
65
66    # 4. Torque penalty (encourage efficient and balanced movements)
67    action_penalty = action_penalty_weight * (torque_penalty + 0.005
          * torque_std_penalty) # Combined action penalties
68
69    # 5. Velocity penalty (discourage high speeds for stability)
70    velocity_penalty = velocity_penalty_weight * (x_vel**2 + y_vel
          **2) # Small penalty proportional to squared velocity
71
72    # Combine all reward components with weights
73    reward = (
74      forward_weight * forward_reward + # Strong encouragement for
            forward movement
75      goal_weight * goal_reward + # Encouragement for reducing
            distance to the goal
76      posture_penalty +          # Penalty for unhealthy posture or
            reward for optimal posture
77      action_penalty +           # Penalize large and uneven torques
78      velocity_penalty           # Penalize excessive velocity
79    )
80
81    return reward
```

Listing 10: Reward function of "v0" dataset *door-human* designed by PROF using $T = 2$.

```python
1 import numpy as np
2
```

```python
def compute_dense_reward(obs: np.ndarray, action: np.ndarray,
        next_obs: np.ndarray) -> float:
    # Extract relevant observations
    latch_angle = next_obs[27] # Latch angular position
    latch_angle_prev = obs[27]
    door_angle = next_obs[28] # Door hinge angular position
    door_angle_prev = obs[28]
    door_open_flag = next_obs[38] # Door open status (1 if open, else
        -1)
    palm_to_handle_dist = np.linalg.norm(next_obs[35:38]) # Distance
        from palm to handle
    palm_to_handle_dist_prev = np.linalg.norm(obs[35:38]) # Previous
        distance from palm to handle

    # Reward weights (parameterized for flexibility)
    latch_progress_weight = 10.0 # Emphasizes latch progress
    door_progress_weight = 15.0 # Emphasizes door progress
    palm_distance_penalty_weight = -1.5 # Penalizes increases in
        distance
    action_penalty_weight = -0.005 # Penalizes large actions
    latch_bonus = 100.0 # Bonus for fully unlocking latch
    door_bonus = 200.0 # Bonus for fully opening door
    intermediate_threshold_bonus = 20.0 # Bonus for crossing
        intermediate thresholds

    # 1. Latch progress reward (potential-based)
    latch_progress = latch_angle - latch_angle_prev
    latch_reward = latch_progress_weight * latch_progress

    # 2. Door progress reward (potential-based)
    door_progress = door_angle - door_angle_prev
    door_reward = door_progress_weight * door_progress

    # 3. Palm-to-handle distance penalty with normalization
    max_distance = np.linalg.norm([1.82, 1.57, 1.57]) # Hypothetical
        max distance
    normalized_distance_penalty = palm_distance_penalty_weight * ((
        palm_to_handle_dist - palm_to_handle_dist_prev) / max_distance
        )

    # 4. Action penalty (normalized)
    action_magnitude = np.sum(action**2) / len(action)
    normalized_action_penalty = action_penalty_weight *
        action_magnitude

    # 5. Bonus rewards for crossing thresholds
    bonus_reward = 0.0
    if latch_angle >= 1.0 and latch_angle_prev < 1.0: # Intermediate
        latch threshold
        bonus_reward += intermediate_threshold_bonus
    if door_angle >= 1.0 and door_angle_prev < 1.0: # Intermediate
        door threshold
        bonus_reward += intermediate_threshold_bonus
    if latch_angle >= 1.82: # Latch fully unlocked
        bonus_reward += latch_bonus * (latch_angle / 1.82) # Scaled
            bonus
    if door_angle >= 1.57 and door_open_flag == 1: # Door fully open
        bonus_reward += door_bonus * (door_angle / 1.57) # Scaled
            bonus

    # 6. Velocity penalty for smoother movements
    latch_velocity = abs(latch_progress)
    velocity_penalty = -0.01 * latch_velocity # Penalizes rapid latch
        movements
```

```
53    # Total reward
54    reward = (
55        latch_reward +
56        door_reward +
57        normalized_distance_penalty +
58        normalized_action_penalty +
59        bonus_reward +
60        velocity_penalty
61    )
62    return reward
```

## G  REWARD FUNCTION CODE CHANGES DURING ITERATION

In this section, we report the optimal reward function code with increasing iterations number $T \in \{0, 1, 2, 3\}$ on representative environments from MuJoCo, AntMaze, and Adroit. The $T = 0$ denotes the initial reward functions without any iterative optimization. Specifically, we use the "v2" dataset `walker2d-medium-replay`, the "v0" datasets `antmaze-medium-diverse` and `door-human` to demonstrate the iteration progression of PROF. The iteration results on `walker2d-medium-replay` are presented in Listing 11, Listing 8, Listing 12 and Listing 13. It can be seen that the optimal reward function at $T = 1$ increases the penalty on excessively rapid oscillations in velocity, joint oscillations, and abrupt changes in actions compared to $T = 0$, leading to improved performance. At $T = 2$, the improved reward function introduces additional penalties applied to both $s$ and $s'$. However, these excessive penalties result in a decline in performance. When $T = 3$, the optimal reward function begins penalizing changes in the z-coordinate and angle of the torso, which are unrelated to the task objectives. Unexpectedly, this reward hack leads to a higher dominance score. Furthermore, the LLM fabricates two input variables, `goal_x` and `prev_action`, which are never provided.

Another example is `antmaze-medium-diverse`, with results in Listing 14, Listing 15, Listing 9 and Listing 16. At $T = 0$ and $T = 1$, the optimal reward function remains the same, indicating that the first iteration did not yield a reward function with a higher dominance score. At $T = 2$, the refined reward function introduces an angle-based reward relative to the target point, scales the goal-reaching reward, adds a penalty on the $z$-coordinate variation of the torso, and penalizes the standard deviation of the actions. These modifications are intuitively beneficial for smooth task completion, and the results in Table 2 confirm their positive impact. After $T = 3$ rounds of optimization, the penalty on the $z$-coordinate of the torso is further strengthened by changing it to a quadratic form. Additionally, noise is added to the forward reward, which intuitively does not facilitate task completion. Although this reward function achieves a higher dominance score, its actual performance declines.

Finally, we analyzed the `door-human` example on Adroit, with results presented in Listing 17, Listing 18, Listing 10 and Listing 19. From $T = 0$ to $T = 1$, the optimal reward function modifies the sub-reward coefficient and introduces an intermediate reward for task completion, leading to improved performance as shown in Table 8. After $T = 2$, the reward function is refined to further enhance performance by scaling the palm-to-handle distance reward and introducing a penalty for rapid latch movements. After $T = 3$, the improved reward function added a penalty for the unchanged angular positions of both the door latch and the door hinge. However, the fabrication of the unprovided `prev_action` and the design of the corresponding penalty resulted in a performance decrease.

In summary, the code-level observations support the conclusions presented in Sections 5.3 and E.2. Specifically, moderate reward optimization enhances performance, whereas excessive optimization induces reward hacking and degrades performance.

Listing 11: Reward function of "v2" dataset *walker2d-medium-replay* designed by PROF using $T = 0$.

```
1  import numpy as np
2
3  def compute_dense_reward(obs: np.ndarray, action: np.ndarray,
       next_obs: np.ndarray) -> float:
4      # Initialize reward
```

```
 5     reward = 0.0
 6
 7     # 1. Forward movement reward (encourage positive x-velocity)
 8     forward_velocity = next_obs[8]
 9     reward += 10.0 * forward_velocity # Weight for forward movement
           reward
10
11     # 2. Posture maintenance
12     # 2.1. Penalize deviation from desired torso height [0.8, 1.0]
13     torso_height = next_obs[0]
14     if torso_height < 0.8:
15         reward -= 10.0 * (0.8 - torso_height) ** 2 # Penalize for
               being too low
16     elif torso_height > 1.0:
17         reward -= 10.0 * (torso_height - 1.0) ** 2 # Penalize for
               being too high
18
19     # 2.2. Penalize deviation from desired torso angle [-1, 1]
20     torso_angle = next_obs[1]
21     if torso_angle < -1.0:
22         reward -= 5.0 * (-1.0 - torso_angle) ** 2 # Penalize for
               extreme backward lean
23     elif torso_angle > 1.0:
24         reward -= 5.0 * (torso_angle - 1.0) ** 2 # Penalize for
               extreme forward lean
25
26     # 3. Action penalty (encourage smooth and efficient movements)
27     reward -= 0.01 * np.sum(np.square(action)) # Small penalty for
           action magnitude
28
29     # 4. Encourage progress (potential-based reward: change in x-
           coordinate)
30     delta_x = next_obs[8] - obs[8] # Change in x-coordinate (progress
           )
31     reward += 2.0 * delta_x # Weight for progress reward
32
33     return reward
```

Listing 12: Reward function of "v2" dataset *walker2d-medium-replay* designed by PROF using $T = 2$.

```
 1 import numpy as np
 2
 3 def compute_dense_reward(obs: np.ndarray, action: np.ndarray,
        next_obs: np.ndarray) -> float:
 4     # Initialize reward
 5     reward = 0.0
 6
 7     # 1. Handle non-finite observations
 8     if not np.isfinite(obs).all() or not np.isfinite(next_obs).all():
 9         return -100.0 # Strong penalty for non-finite states
10
11     # 2. Forward movement reward (primary goal)
12     forward_velocity = next_obs[8]
13     reward += 5.0 * forward_velocity # Weight for forward movement
           reward
14
15     # 3. Progress reward (potential-based reward with efficiency
           considerations)
16     delta_x = next_obs[8] - obs[8] # Change in x-coordinate (progress
           )
17     reward += 4.0 * delta_x # Adjusted weight for progress reward to
           better balance it with forward velocity
18
19     # 4. Posture maintenance
```

```
20      # 4.1. Penalize deviation from desired torso height [0.8, 1.0]
21      torso_height = next_obs[0]
22      if torso_height < 0.8:
23          reward -= 5.0 * (0.8 - torso_height) ** 2 # Penalize for being
                  too low
24      elif torso_height > 1.0:
25          reward -= 5.0 * (torso_height - 1.0) ** 2 # Penalize for being
                  too high
26
27      # Penalize torso height deviations in the current state as well
28      if obs[0] < 0.8:
29          reward -= 2.5 * (0.8 - obs[0]) ** 2
30      elif obs[0] > 1.0:
31          reward -= 2.5 * (obs[0] - 1.0) ** 2
32
33      # 4.2. Penalize deviation from desired torso angle [-1, 1]
34      torso_angle = next_obs[1]
35      if torso_angle < -1.0:
36          reward -= 3.0 * (-1.0 - torso_angle) ** 2 # Penalize for
                  extreme backward lean
37      elif torso_angle > 1.0:
38          reward -= 3.0 * (torso_angle - 1.0) ** 2 # Penalize for
                  extreme forward lean
39
40      # Penalize torso angle deviations in the current state as well
41      if obs[1] < -1.0:
42          reward -= 1.5 * (-1.0 - obs[1]) ** 2
43      elif obs[1] > 1.0:
44          reward -= 1.5 * (obs[1] - 1.0) ** 2
45
46      # 5. Action penalty (encourage smooth and efficient movements)
47      reward -= 0.05 * np.sum(np.square(action)) # Weight for action
            penalty to discourage excessive torque
48
49      # 6. Refine smooth progress and velocity penalties
50      velocity_change_penalty = 0.5 * np.abs(next_obs[8] - obs[8]) #
            Penalize abrupt velocity changes
51      reward -= velocity_change_penalty
52
53      # 7. Penalize unsafe behaviors
54      joint_velocity_penalty = 0.01 * np.sum(np.abs(next_obs[10:])) #
            Penalize rapid joint oscillations
55      action_smoothness_penalty = 0.01 * np.sum(np.abs(action - np.mean
            (action))) # Penalize abrupt changes in actions
56      reward -= joint_velocity_penalty + action_smoothness_penalty
57
58      # 8. Encourage exploration and robustness
59      if forward_velocity > 2.0:
60          reward += 2.0 # Small bonus for achieving higher forward
                  velocity
61      if delta_x > 1.0:
62          reward += 1.0 # Small bonus for sustained progress
63
64      return reward
```

Listing 13: Reward function of "v2" dataset *walker2d-medium-replay* designed by PROF using $T = 3$.

```
1  import numpy as np
2
3  def compute_dense_reward(obs: np.ndarray, action: np.ndarray,
       next_obs: np.ndarray, goal_x: float = None, prev_action: np.
       ndarray = None) -> float:
4      """
5      Compute the dense reward for the agent.
```

```
 6
 7    Args:
 8        obs (np.ndarray): Current observation.
 9        action (np.ndarray): Current action.
10        next_obs (np.ndarray): Next observation.
11        goal_x (float, optional): Target x-coordinate for potential-
              based reward. Defaults to None.
12        prev_action (np.ndarray, optional): Previous action for
              smoothness penalty. Defaults to None.
13
14    Returns:
15        float: Computed dense reward.
16    """
17    # Initialize reward
18    reward = 0.0
19
20    # 1. Handle non-finite observations
21    if not np.isfinite(obs).all() or not np.isfinite(next_obs).all():
22        return -100.0 # Strong penalty for non-finite states
23
24    # 2. Forward movement reward (primary goal)
25    forward_velocity = next_obs[8]
26    reward += 5.0 * forward_velocity # Primary reward for forward
          movement
27
28    # 3. Progress reward (potential-based reward with efficiency
          considerations)
29    delta_x = next_obs[8] - obs[8] # Change in x-coordinate (progress
          )
30    reward += 2.0 * delta_x # Adjusted weight for progress reward to
          balance with forward velocity
31
32    # Optional target-based potential reward
33    if goal_x is not None:
34        distance_to_goal_prev = abs(obs[8] - goal_x)
35        distance_to_goal_next = abs(next_obs[8] - goal_x)
36        reward += 3.0 * (distance_to_goal_prev - distance_to_goal_next
              ) # Reward for reducing distance to goal
37
38    # 4. Posture maintenance
39    # 4.1. Penalize deviation from desired torso height [0.8, 1.0]
40    torso_height = next_obs[0]
41    if torso_height < 0.8:
42        reward -= 5.0 * (0.8 - torso_height) ** 2 # Penalize for being
               too low
43    elif torso_height > 1.0:
44        reward -= 5.0 * (torso_height - 1.0) ** 2 # Penalize for being
               too high
45
46    # Penalize torso height deviations in the current state as well
47    if obs[0] < 0.8:
48        reward -= 2.5 * (0.8 - obs[0]) ** 2
49    elif obs[0] > 1.0:
50        reward -= 2.5 * (obs[0] - 1.0) ** 2
51
52    # 4.2. Penalize deviation from desired torso angle [-1, 1]
53    torso_angle = next_obs[1]
54    if torso_angle < -1.0:
55        reward -= 3.0 * (-1.0 - torso_angle) ** 2 # Penalize for
              extreme backward lean
56    elif torso_angle > 1.0:
57        reward -= 3.0 * (torso_angle - 1.0) ** 2 # Penalize for
              extreme forward lean
58
59    # Penalize torso angle deviations in the current state as well
```

```
60     if obs[1] < -1.0:
61         reward -= 1.5 * (-1.0 - obs[1]) ** 2
62     elif obs[1] > 1.0:
63         reward -= 1.5 * (obs[1] - 1.0) ** 2
64
65     # Penalize posture dynamics over time
66     reward -= 1.0 * (abs(next_obs[0] - obs[0]) + abs(next_obs[1] -
           obs[1])) # Penalize large posture changes
67
68     # 5. Action penalty (encourage smooth and efficient movements)
69     reward -= 0.05 * np.sum(np.square(action)) # Penalize large
           torque values
70
71     # Penalize abrupt changes in actions
72     if prev_action is not None:
73         reward -= 0.01 * np.sum(np.abs(action - prev_action)) #
               Penalize abrupt action changes
74
75     # 6. Refine smooth progress and velocity penalties
76     acceleration = abs(next_obs[8] - obs[8]) # Measure acceleration
77     reward -= 0.5 * acceleration # Penalize large accelerations
78
79     # Penalize rapid joint oscillations
80     joint_velocity_penalty = 0.01 * np.sum(np.abs(next_obs[10:]))
81     reward -= joint_velocity_penalty
82
83     # 7. Encourage exploration and robustness
84     if forward_velocity > 2.0:
85         reward += 2.0 + 0.5 * (forward_velocity - 2.0) # Dynamic bonus
               for higher forward velocity
86     if delta_x > 1.0:
87         reward += 1.0 + 0.2 * (delta_x - 1.0) # Dynamic bonus for
               sustained progress
88
89     return reward
```

Listing 14: Reward function of "v0" dataset *antmaze-medium-diverse* designed by PROF using $T = 0$.

```
1  import numpy as np
2
3  def compute_dense_reward(obs: np.ndarray, action: np.ndarray,
        next_obs: np.ndarray) -> float:
4      # Extract relevant variables from the observations
5      x_pos, y_pos, z_pos = obs[0], obs[1], obs[2] # Current position
6      next_x_pos, next_y_pos, next_z_pos = next_obs[0], next_obs[1],
            next_obs[2] # Next position
7      x_vel, y_vel = obs[15], obs[16] # Current velocities
8      goal_x, goal_y = obs[29], obs[30] # Goal position
9
10     # Extract actions for penalty
11     torque_penalty = np.sum(np.square(action)) # Sum of squared
            torques (penalize large actions)
12
13     # Compute distances to the goal
14     current_goal_dist = np.sqrt((goal_x - x_pos)**2 + (goal_y - y_pos
            )**2) # Current distance to goal
15     next_goal_dist = np.sqrt((goal_x - next_x_pos)**2 + (goal_y -
            next_y_pos)**2) # Next distance to goal
16
17     # Reward components
18     # 1. Directional reward for moving forward
19     forward_reward = (next_x_pos - x_pos) + (next_y_pos - y_pos) #
            Positive displacement in x and y
20
```

```
21      # 2. Goal-reaching reward (potential-based reward: reduction in
            distance to goal)
22      goal_reward = current_goal_dist - next_goal_dist # Reward for
            reducing distance to the goal
23
24      # 3. Posture penalty (encourage healthy z-pos in range [0.2,
            1.0])
25      if next_z_pos < 0.2 or next_z_pos > 1.0 or not np.isfinite(
            next_z_pos):
26          posture_penalty = -10.0 # Strong penalty for unhealthy posture
27      else:
28          posture_penalty = 0.0 # No penalty for healthy posture
29
30      # 4. Torque penalty (encourage efficient movements)
31      action_penalty = -0.01 * torque_penalty # Small penalty for large
             torques
32
33      # 5. Velocity penalty (discourage high speeds for stability)
34      velocity_penalty = -0.005 * (x_vel**2 + y_vel**2) # Small penalty
             proportional to squared velocity
35
36      # Combine all reward components with weights
37      reward = (
38          5.0 * forward_reward + # Strong encouragement for forward
                 movement
39          3.0 * goal_reward + # Encouragement for reducing distance to
                 the goal
40          posture_penalty + # Penalty for unhealthy posture
41          action_penalty +  # Penalize large torques
42          velocity_penalty  # Penalize excessive velocity
43      )
44
45      return reward
```

Listing 15: Reward function of "v0" dataset *antmaze-medium-diverse* designed by PROF using $T = 1$.

```
1   import numpy as np
2
3   def compute_dense_reward(obs: np.ndarray, action: np.ndarray,
        next_obs: np.ndarray) -> float:
4       # Extract relevant variables from the observations
5       x_pos, y_pos, z_pos = obs[0], obs[1], obs[2] # Current position
6       next_x_pos, next_y_pos, next_z_pos = next_obs[0], next_obs[1],
            next_obs[2] # Next position
7       x_vel, y_vel = obs[15], obs[16] # Current velocities
8       goal_x, goal_y = obs[29], obs[30] # Goal position
9
10      # Extract actions for penalty
11      torque_penalty = np.sum(np.square(action)) # Sum of squared
            torques (penalize large actions)
12
13      # Compute distances to the goal
14      current_goal_dist = np.sqrt((goal_x - x_pos)**2 + (goal_y - y_pos
            )**2) # Current distance to goal
15      next_goal_dist = np.sqrt((goal_x - next_x_pos)**2 + (goal_y -
            next_y_pos)**2) # Next distance to goal
16
17      # Reward components
18      # 1. Directional reward for moving forward
19      forward_reward = (next_x_pos - x_pos) + (next_y_pos - y_pos) #
            Positive displacement in x and y
20
21      # 2. Goal-reaching reward (potential-based reward: reduction in
            distance to goal)
```

```
22    goal_reward = current_goal_dist - next_goal_dist # Reward for
          reducing distance to the goal
23
24    # 3. Posture penalty (encourage healthy z-pos in range [0.2,
          1.0])
25    if next_z_pos < 0.2 or next_z_pos > 1.0 or not np.isfinite(
          next_z_pos):
26        posture_penalty = -10.0 # Strong penalty for unhealthy posture
27    else:
28        posture_penalty = 0.0 # No penalty for healthy posture
29
30    # 4. Torque penalty (encourage efficient movements)
31    action_penalty = -0.01 * torque_penalty # Small penalty for large
          torques
32
33    # 5. Velocity penalty (discourage high speeds for stability)
34    velocity_penalty = -0.005 * (x_vel**2 + y_vel**2) # Small penalty
          proportional to squared velocity
35
36    # Combine all reward components with weights
37    reward = (
38        5.0 * forward_reward + # Strong encouragement for forward
              movement
39        3.0 * goal_reward + # Encouragement for reducing distance to
              the goal
40        posture_penalty + # Penalty for unhealthy posture
41        action_penalty +  # Penalize large torques
42        velocity_penalty  # Penalize excessive velocity
43    )
44
45    return reward
```

Listing 16: Reward function of "v0" dataset *antmaze-medium-diverse* designed by PROF using $T = 3$.

```
1  import numpy as np
2
3  def compute_dense_reward(
4      obs: np.ndarray,
5      action: np.ndarray,
6      next_obs: np.ndarray,
7      forward_weight: float = 5.0,
8      goal_weight: float = 3.0,
9      posture_penalty_weight: float = -10.0,
10     action_penalty_weight: float = -0.01,
11     velocity_penalty_weight: float = -0.005,
12     z_target: float = 0.6,
13     z_tolerance: float = 0.01,
14     velocity_clip: float = 10.0,
15     exploration_noise: float = 0.1
16 ) -> float:
17     """
18     Computes the dense reward for the RL environment, considering
            progress toward the goal,
19     efficient movements, healthy posture, and stability.
20
21     Args:
22         obs (np.ndarray): Current observation.
23         action (np.ndarray): Action taken.
24         next_obs (np.ndarray): Next observation.
25         forward_weight (float): Weight for the forward movement reward
                .
26         goal_weight (float): Weight for the goal-reaching reward.
27         posture_penalty_weight (float): Weight for the posture penalty
                .
```

```
28          action_penalty_weight (float): Weight for the action penalty.
29          velocity_penalty_weight (float): Weight for the velocity
                penalty.
30          z_target (float): Target height for the torso.
31          z_tolerance (float): Tolerance for the posture penalty.
32          velocity_clip (float): Maximum velocity value for clipping.
33          exploration_noise (float): Noise factor to encourage
                exploration.
34
35      Returns:
36          float: The computed reward.
37      """
38      # Extract relevant variables from the observations
39      x_pos, y_pos, z_pos = obs[0], obs[1], obs[2] # Current position
40      next_x_pos, next_y_pos, next_z_pos = next_obs[0], next_obs[1],
            next_obs[2] # Next position
41      x_vel, y_vel = np.clip(obs[15], -velocity_clip, velocity_clip),
            np.clip(obs[16], -velocity_clip, velocity_clip) # Clipped
            velocities
42      goal_x, goal_y = obs[29], obs[30] # Goal position
43
44      # Extract actions for penalty
45      torque_penalty = np.sum(np.square(action)) # Sum of squared
            torques (penalize large actions)
46      torque_std_penalty = np.std(action) # Penalize uneven torque
            application
47
48      # Compute distances to the goal
49      current_goal_dist = np.sqrt((goal_x - x_pos)**2 + (goal_y - y_pos
            )**2) # Current distance to goal
50      next_goal_dist = np.sqrt((goal_x - next_x_pos)**2 + (goal_y -
            next_y_pos)**2) # Next distance to goal
51
52      # Reward components
53      # 1. Directional reward for moving toward the goal
54      goal_direction = np.array([goal_x - x_pos, goal_y - y_pos])
55      if np.linalg.norm(goal_direction) > 0:
56          goal_direction = goal_direction / np.linalg.norm(
                goal_direction) # Normalize the goal direction
57      movement_vector = np.array([next_x_pos - x_pos, next_y_pos -
            y_pos])
58      forward_reward = np.dot(movement_vector, goal_direction) # Reward
             for moving in the desired direction
59
60      # Add exploration noise for early learning stages
61      forward_reward += np.random.uniform(-exploration_noise,
            exploration_noise)
62
63      # 2. Goal-reaching reward (potential-based reward: reduction in
            distance to goal)
64      goal_reward = ((current_goal_dist - next_goal_dist) / max(
            current_goal_dist, 1e-8)) if current_goal_dist > 0 else 0.0
65
66      # 3. Posture penalty (encourage healthy z-pos in range [0.2,
            1.0])
67      if next_z_pos < (0.2 - z_tolerance) or next_z_pos > (1.0 +
            z_tolerance):
68          posture_penalty = posture_penalty_weight # Strong penalty for
                 unhealthy posture
69      else:
70          posture_penalty = -0.5 * (next_z_pos - z_target)**2 #
                Quadratic penalty for deviations from target height
71
72      # 4. Torque penalty (encourage efficient and balanced movements)
```

```
73    action_penalty = action_penalty_weight * (torque_penalty + 0.005
          * torque_std_penalty) # Combined action penalties
74
75    # 5. Velocity penalty (discourage high speeds for stability)
76    velocity_penalty = velocity_penalty_weight * (x_vel**2 + y_vel
          **2) # Small penalty proportional to squared velocity
77
78    # Combine all reward components with weights
79    reward = (
80        forward_weight * forward_reward + # Strong encouragement for
              forward movement
81        goal_weight * goal_reward + # Encouragement for reducing
              distance to the goal
82        posture_penalty +          # Penalty for unhealthy posture or
              reward for optimal posture
83        action_penalty +           # Penalize large and uneven torques
84        velocity_penalty           # Penalize excessive velocity
85    )
86
87    # Ensure reward is finite and within a reasonable range
88    if not np.isfinite(reward):
89        reward = -100.0 # Heavy penalty for non-finite rewards
90
91    return np.clip(reward, -1.0, 1.0) # Normalize reward to [-1, 1]
```

Listing 17: Reward function of "v0" dataset *door-human* designed by PROF using $T = 0$.

```
1  import numpy as np
2
3  def compute_dense_reward(obs: np.ndarray, action: np.ndarray,
       next_obs: np.ndarray) -> float:
4      # Extract relevant observations
5      latch_angle = next_obs[27] # Latch angular position
6      latch_angle_prev = obs[27]
7      door_angle = next_obs[28] # Door hinge angular position
8      door_angle_prev = obs[28]
9      door_open_flag = next_obs[38] # Door open status (1 if open, else
           -1)
10     palm_to_handle_dist = np.linalg.norm(next_obs[35:38]) # Distance
            from palm to handle
11
12     # Reward weights
13     latch_progress_weight = 5.0
14     door_progress_weight = 10.0
15     palm_distance_penalty_weight = -2.0
16     action_penalty_weight = -0.01
17     latch_bonus = 100.0
18     door_bonus = 200.0
19
20     # 1. Latch progress reward
21     latch_progress = latch_angle - latch_angle_prev
22     latch_reward = latch_progress * latch_progress_weight
23
24     # 2. Door progress reward
25     door_progress = door_angle - door_angle_prev
26     door_reward = door_progress * door_progress_weight
27
28     # 3. Palm-to-handle distance penalty
29     distance_penalty = palm_distance_penalty_weight *
           palm_to_handle_dist
30
31     # 4. Action penalty
32     action_penalty = action_penalty_weight * np.sum(action**2)
33
34     # 5. Bonus rewards for crossing thresholds
```

```
35    bonus_reward = 0.0
36    if latch_angle >= 1.82: # Latch fully unlocked
37        bonus_reward += latch_bonus
38    if door_angle >= 1.57 and door_open_flag == 1: # Door fully open
39        bonus_reward += door_bonus
40
41    # Total reward
42    reward = latch_reward + door_reward + distance_penalty +
             action_penalty + bonus_reward
43    return reward
```

Listing 18: Reward function of "v0" dataset *door-human* designed by PROF using $T = 1$.

```
1  import numpy as np
2
3  def compute_dense_reward(obs: np.ndarray, action: np.ndarray,
      next_obs: np.ndarray) -> float:
4      # Extract relevant observations
5      latch_angle = next_obs[27] # Latch angular position
6      latch_angle_prev = obs[27]
7      door_angle = next_obs[28] # Door hinge angular position
8      door_angle_prev = obs[28]
9      door_open_flag = next_obs[38] # Door open status (1 if open, else
          -1)
10     palm_to_handle_dist = np.linalg.norm(next_obs[35:38]) # Distance
          from palm to handle
11     palm_to_handle_dist_prev = np.linalg.norm(obs[35:38]) # Previous
          distance from palm to handle
12
13     # Reward weights
14     latch_progress_weight = 10.0
15     door_progress_weight = 15.0
16     palm_distance_penalty_weight = -1.5
17     action_penalty_weight = -0.005
18     latch_bonus = 100.0
19     door_bonus = 200.0
20     intermediate_threshold_bonus = 20.0
21
22     # 1. Latch progress reward (potential-based)
23     latch_progress = latch_angle - latch_angle_prev
24     latch_reward = latch_progress_weight * latch_progress
25
26     # 2. Door progress reward (potential-based)
27     door_progress = door_angle - door_angle_prev
28     door_reward = door_progress_weight * door_progress
29
30     # 3. Palm-to-handle distance penalty with positive reinforcement
          for reduction
31     distance_penalty = palm_distance_penalty_weight * (
          palm_to_handle_dist - palm_to_handle_dist_prev)
32
33     # 4. Action penalty (state-dependent)
34     action_magnitude = np.sum(action**2)
35     action_penalty = action_penalty_weight * action_magnitude
36
37     # 5. Bonus rewards for crossing thresholds
38     bonus_reward = 0.0
39     if latch_angle >= 1.0 and latch_angle_prev < 1.0: # Intermediate
          latch threshold
40         bonus_reward += intermediate_threshold_bonus
41     if door_angle >= 1.0 and door_angle_prev < 1.0: # Intermediate
          door threshold
42         bonus_reward += intermediate_threshold_bonus
43     if latch_angle >= 1.82: # Latch fully unlocked
44         bonus_reward += latch_bonus
```

```
45      if door_angle >= 1.57 and door_open_flag == 1: # Door fully open
46          bonus_reward += door_bonus
47
48      # Total reward
49      reward = latch_reward + door_reward + distance_penalty +
            action_penalty + bonus_reward
50      return reward
```

Listing 19: Reward function of "v0" dataset *door-human* designed by PROF using $T = 3$.

```
1  import numpy as np
2
3  def compute_dense_reward(obs: np.ndarray, action: np.ndarray,
       next_obs: np.ndarray) -> float:
4      # Extract relevant observations
5      latch_angle = next_obs[27] # Latch angular position
6      latch_angle_prev = obs[27]
7      door_angle = next_obs[28] # Door hinge angular position
8      door_angle_prev = obs[28]
9      door_open_flag = next_obs[38] # Door open status (1 if open, else
           -1)
10     palm_to_handle_dist = np.linalg.norm(next_obs[35:38]) # Distance
           from palm to handle
11     palm_to_handle_dist_prev = np.linalg.norm(obs[35:38]) # Previous
           distance from palm to handle
12
13     # Reward weights (parameterized for flexibility)
14     latch_progress_weight = 10.0 # Emphasizes latch progress
15     door_progress_weight = 15.0 # Emphasizes door progress
16     palm_distance_penalty_weight = -1.5 # Penalizes increases in
           distance
17     action_penalty_weight = -0.005 # Penalizes large actions
18     latch_bonus = 100.0 # Bonus for fully unlocking latch
19     door_bonus = 200.0 # Bonus for fully opening door
20     intermediate_threshold_bonus = 20.0 # Bonus for crossing
           intermediate thresholds
21
22     # 1. Normalize progress rewards
23     latch_progress = (latch_angle - latch_angle_prev) / 1.82 #
           Normalized by latch max range
24     latch_reward = latch_progress_weight * latch_progress
25
26     door_progress = (door_angle - door_angle_prev) / 1.57 #
           Normalized by door max range
27     door_reward = door_progress_weight * door_progress
28
29     # 2. Palm-to-handle distance penalty with dynamic normalization
30     max_distance = np.linalg.norm([1.82, 1.57, 1.57]) # Hypothetical
           max distance
31     normalized_distance_penalty = palm_distance_penalty_weight * (
32         (palm_to_handle_dist - palm_to_handle_dist_prev) /
               max_distance
33     )
34
35     # 3. Action penalty (normalized)
36     action_magnitude = np.sum(action**2) / len(action)
37     normalized_action_penalty = action_penalty_weight *
           action_magnitude
38
39     # 4. Bonus rewards for crossing thresholds (scaled)
40     bonus_reward = 0.0
41     if latch_angle >= 1.0 and latch_angle_prev < 1.0: # Intermediate
           latch threshold
42         bonus_reward += intermediate_threshold_bonus * (latch_angle /
               1.0)
```

```
43     if door_angle >= 1.0 and door_angle_prev < 1.0: # Intermediate
            door threshold
44         bonus_reward += intermediate_threshold_bonus * (door_angle /
            1.0)
45     if latch_angle >= 1.82: # Latch fully unlocked
46         bonus_reward += latch_bonus * (latch_angle / 1.82)
47     if door_angle >= 1.57 and door_open_flag == 1: # Door fully open
48         bonus_reward += door_bonus * (door_angle / 1.57)
49
50     # 5. Velocity penalty for smoother movements
51     max_latch_velocity = 1.82 / 10.0 # Hypothetical max change in
            latch position
52     latch_velocity = abs(latch_angle - latch_angle_prev) /
            max_latch_velocity
53     velocity_penalty = -0.01 * latch_velocity # Penalizes rapid latch
            movements
54
55     # 6. Penalty for abrupt action changes
56     prev_action = np.zeros_like(action) # Placeholder for previous
            actions (use actual if available)
57     action_smoothness_penalty = -0.01 * np.linalg.norm(action -
            prev_action) # Encourages smoother actions
58
59     # 7. Stagnation penalty for lack of progress
60     stagnation_penalty = -5.0 if abs(latch_progress) < 0.01 and abs(
            door_progress) < 0.01 else 0.0
61
62     # Total reward
63     reward = (
64         latch_reward +
65         door_reward +
66         normalized_distance_penalty +
67         normalized_action_penalty +
68         bonus_reward +
69         velocity_penalty +
70         action_smoothness_penalty +
71         stagnation_penalty
72     )
73
74     return reward
```

