# OpenReview forum: "PROF: An LLM-based Reward Code Preference Optimization Framework for Offline Imitation Learning"
_ICLR.cc/2026/Conference — ICLR 2026 Conference Withdrawn Submission_

### Official Review · Reviewer_y2AM · 2025-10-29

**Soundness:** 2
**Presentation:** 3
**Contribution:** 2
**Rating:** 4
**Confidence:** 3

**Summary:**

In this paper, a framework, namely PROF, is proposed by leveraging LLMs to generate and improve executable reward function codes from natural language descriptions and a single expert trajectory. In which, Reward Preference Ranking (RPR) is introduced for the reward quality assessment and ranking, then text-based gradient optimization is utilized to automate the selection and refinement of optimal reward functions for downstream policy learning. Experimental results conducted on D4RL demonstrated the effectiveness of the presented approach.

**Strengths:**

1. By leveraging LLMs to generate and improve executable reward function codes is promising, especially for the offline RL setting.
2. Experimental results show that PROF achieves similar or better performance against some baselines on the D4RL benchmark.
3. The integration of LLM, preference ranking, and textual optimization for addressing the reward design issue is novel and effective.

**Weaknesses:**

1. While experimental results on some cases demonstrate the effectiveness of the presented framework, theoretical guarantees are lacking for the performance improvements.
2. In the paper, only one expert trajectory is utilized, which is in contrast to reality, where optimal behaviors may be diverse and not necessarily proximal to a limited set of demonstrations (as also claimed in the paper). Experiments with multiple expert demonstrations are suggested.
3. With the help of LLM, some prior knowledge is incorporated for the presented framework, but for some baselines, such knowledge is not available, the performance comparison in the experiments in some sense is unfair.
4. The most recent baseline for the comparison is publicly available in 2023 (seabo), more recent baselines or existing methods but can also leverage LLM to provide some prior knowledge should be compared and discussed.

**Questions:**

Please refer to the weakness points.

---

### Official Review · Reviewer_HDdH · 2025-10-29

**Soundness:** 3
**Presentation:** 2
**Contribution:** 1
**Rating:** 2
**Confidence:** 4

**Summary:**

The authors introduce a method for offline imitation learning in scenarios where both expert and non-expert trajectories are available. The core idea is to generate a set of reward functions by prompting a large language model (LLM) with a description of a Gym environment. The method then selects the best reward function via preference ranking and performs a text-gradient update to further refine it. The authors evaluate their approach on several baselines, including the D4RL dataset.

**Strengths:**

The authors evaluated the algorithm on multiple environments as well as using multiple LLMs.

**Weaknesses:**

1. The authors compare their method only against non–state-of-the-art algorithms [1,2] in this field. When considering more recent work in this area [1,2], the proposed method would likely underperform. Therefore, the claim of outperforming all baselines appears to be based on a selective choice of comparisons, which gives the impression of cherry-picking baselines to favor their approach.

2. The authors mention that a single iteration is sufficient for full training. In that case, how does this approach compare to Eureka [3]? The two methods seem conceptually similar, except that the proposed method uses textgrad instead of LLM-based reflection.

[1,2] https://arxiv.org/abs/2402.13037, https://arxiv.org/pdf/2507.12815
[3]    https://arxiv.org/pdf/2310.12931

**Questions:**

1. Could you provide additional benchmark evaluations against actual SOTA methods?
2. Could you comment on how your paper is different from eureka [3]?
3. What are the potential limitations of the method?

---

### Official Review · Reviewer_zKh7 · 2025-11-01

**Soundness:** 3
**Presentation:** 2
**Contribution:** 2
**Rating:** 2
**Confidence:** 4

**Summary:**

This paper introduces a framework, PROF, that uses LLMs to automatically generate and optimize reward function code for imitation learning. Instead of relying on prelabeled rewards or environment interactions, it produces candidate reward functions from textual prompts and refines them through an iterative "Reward Preference Ranking" process, which evaluates reward quality using expert trajectories and unlabeled data. The framework then applies TextGrad, a textual gradient optimization method, to iteratively improve the reward code. Experiments on D4RL show that PROF consistently matches or outperforms the baselines across multiple domains.

**Strengths:**

- The paper automates reward function generation and optimization using LLMs without requiring environment interaction.
- It achieves strong empirical performance, surpassing or matching baselines across D4RL tasks.

**Weaknesses:**

Offline algorithms make sense when interactions with the environment is costly. Here, to prompt the LLM for the reward functions, the environment information is fed into LLM. In the examples in the paper, this seems to be just the observation and action space (according to Appendix C.2), which is fine. But I am not sure if this amount of information would be sufficient in more realistic and complex environments. It may be needed to input the dynamics as well (e.g., step function in the gym convention). But that would mean there is already a simulator. If there is a simulator, online algorithms are not bad at all. I think the paper should demonstrate performance on a real robot environment to make their argument convincing.

While this is my most major concern, below are more minor comments:
- In line 103, should it be "IL" instead of "RL"? Because BC is not an offline RL algorithm.
- "Reward Design via LLMs" subsection may want to talk about method that use LLMs or VLMs to generate/learn rewards (though VLMs may be less relevant here). Some examples are: RoboCLIP, RLAIF, RL-VLM-F, Video2Reward.
- Equation 4 is confusing in that it made me think the paper assumes access to the reward values of the expert demonstrations. This is not the case, but I find the presentation is confusing in general. It makes it difficult for the reader that the paper first defines everything without explaining why and then the next subsection tells the method/intuition.
- Injecting Gaussian noise to the demonstrations (especially the actions) reminds me of the D-REX paper by Brown et al. It should be cited.
- Equation 5 computes a standard deviation over a set of vectors. While the meaning is clear, this is not mathematically well-defined. I recommend that the paper should stick to formal math notation when introducing something mathematically.
- Similarly, Equation 7 defines noise-injected observations and actions as "vector plus distribution," but mathematically, this is a distribution, and not a sample from that distribution.
- The paper should include a comparison against Eureka (Ma et al.) and its variants. That is an important baseline for this work. It can be argued that it is online RL, but it does not have to implemented online. It can be applied on an offline RL setting as well.

**Questions:**

See the questions and the comments in the weaknesses section.

---

### Official Review · Reviewer_6aP8 · 2025-11-02

**Soundness:** 3
**Presentation:** 3
**Contribution:** 3
**Rating:** 6
**Confidence:** 3

**Summary:**

This paper introduces **PROF**, an LLM-based framework for *offline imitation learning* that learns reward functions from one expert trajectory and unlabeled data. PROF first uses an LLM to generate executable reward code, then applies a *Reward Preference Ranking* (RPR) score to rank candidates without environment interaction, and finally refines them with TextGrad-style code optimization. Experiments on D4RL MuJoCo, AntMaze, and Adroit show consistent gains, especially on AntMaze and Adroit, with comparisons to IQL, SEABO, and OTR. The method achieves competitive or superior results under fair evaluation with five seeds and standard deviations reported.

**Strengths:**

* **Originality:** Introduces a formal reward-ranking signal (RPR) that does not require environment interaction; conceptually novel for offline IL.
* **Empirical quality:** Evaluated on three diverse D4RL domains with multiple seeds and std devs reported. Competitive or superior to recent baselines (SEABO, OTR) under standard normalized-score protocols.
* **Transparency:** Provides clear algorithmic structure (generation → ranking → optimization) and example prompts; includes iteration ablations and LLM comparisons.
* **Significance:** Meaningful practical impact for offline imitation learning; demonstrates reward generalization across domains and partial LLM portability.
* **Fairness:** Baselines include recent state-of-the-art (SEABO 2024, OTR 2023), evaluated in the same framework.

**Weaknesses:**

1. **Missing component ablations:** No sensitivity study for key RPR hyperparameters $(\delta, \alpha_o, \alpha_a, H)$ or the noise-perturbation mechanism. This limits understanding of what drives performance.
2. **Reward hacking risk:** At iteration $T=3$, the LLM fabricates variables, leading to spurious but high-ranked rewards. Indicates misalignment between ranking score and real performance.
3. **Baseline parity unclear:** Baseline results (SEABO, OTR) seem taken from prior work rather than fully re-run under identical conditions. Cross-paper reuse may bias comparisons.
4. **Limited generality:** PROF is mostly tested with IQL; only one TD3+BC example is shown. Generality across learners (e.g., CQL, AWAC) remains untested.
5. **Scaling heuristics:** Task-specific reward scaling (Table 7) may reduce claims of plug-and-play reward generalization.
6. **Narrow LLM ablation:** Cross-LLM results only on HalfCheetah; unclear whether findings hold for more complex tasks.
7. **Related work positioning:** Needs clearer distinction from recent online LLM-based reward design works (e.g., CARD) to sharpen the novelty claim.

**Questions:**

1. How sensitive is RPR to $(\delta, \alpha_o, \alpha_a, H)$? Please provide ablations.
2. How can the method detect or mitigate reward hacking during later iterations (e.g., fabricated variables)?
3. Were SEABO and OTR re-run in your environment with identical seeds and budgets?
4. Does PROF generalize to other offline RL algorithms (CQL, AWAC)?
5. Can cross-LLM tests be replicated on AntMaze and Adroit to confirm portability?
6. Were the reward-scaling heuristics tuned per task? What is the performance change without them?
7. Will full code, prompts, and seeds be released?

---

### Note · Authors · 2025-11-14

I have read and agree with the venue's withdrawal policy on behalf of myself and my co-authors.